

**Title:** **Quantifying memory and persistence in the atmosphere–land/ocean carbon system**

**Authors:** **Matthias Jonas[1*],**

**Rostyslav Bun[2,3],**

**Iryna Ryzha[2] &**

**Piotr Żebrowski[1]**

[1]Advancing Systems Analysis Program, International Institute for Applied Systems Analysis, 2361, Laxenburg, Austria. [2]Department of Applied Mathematics, Lviv Polytechnic National University, 79013, Lviv, Ukraine. [3]Department of Transport and Computer Sciences, WSB University, 41300, Dąbrowa Górnicza, Poland. [*]email: jonas@iiasa.ac.at

Keywords: Global carbon cycle, global atmosphere–land/ocean system, atmospheric $CO_2$ emissions, stress-strain model, Maxwell body, memory, persistence




**Abstract**
Here we interpret carbon dioxide ($CO_2$) emissions from fossil fuel burning and land use as a
global stress-strain experiment. We use the idea of a Maxwell body consisting of elastic and
damping (viscous) elements to reflect the overall behaviour of the atmosphere–land/ocean
system in response to the continued increase of $CO_2$ emissions between 1850 and 2015. From
the standpoint of a global observer, we see that as a consequence of the increase, the $CO_2$
concentration in the atmosphere increases (rather quickly). Concomitantly, the atmosphere
warms and expands, while part of the carbon is locked away (rather slowly) in land and
oceans, likewise under the influence of global warming.
It is not known how reversible and how much out of sync the latter process is in relation to
the former. All we know is that the slower process remembers the influence of the faster one
which runs ahead. Here we ask three (nontrivial) questions: (1) Can this global-scale
memory—Earth's memory—be quantified? (2) Is Earth's memory a buffer which is
negligently exploited; and in the case that it is even a limited buffer, what is the degree of
exploitation? And (3) does Earth's memory allow its persistence (path dependency) to be
quantified? To the best of our knowledge, the answers to these questions are pending.
We go beyond textbook knowledge by introducing three parameters that characterise the
system: delay time, memory, and persistence. The three parameters depend, ceteris paribus,
solely on the system's characteristic viscoelastic behaviour and allow deeper insights into that
system. We find that since 1850, the atmosphere–land/ocean system has been trapped
progressively in terms of persistence (i.e., it will become progressively more difficult to
strain-relax the system), while its ability to build up memory has been reduced. The ability of
a system to build up memory effectively can be understood as its ability to respond still





within its natural regime; or, if the build-up of memory is limited, as a measure for system
failures globally in the future. Approximately 60% of Earth's memory had already been
exploited by humankind prior to 1959.  We expect system failures globally well before 2050
if the current trend in emissions is not reversed immediately and sustainably.



## 1. Motivation

Over the last century anthropogenic pressure on Earth became increasingly noticeable.
Human activities turned out to be so pervasive and profound that the very life support system
upon which humans depend is threatened (Steffen et al., 2004, 2015). The increase of
emissions of greenhouse gases into the atmosphere is only one of several serious global
threats and their reduction is in the center of international agreements (Steffen et al., 2015;
United Nations, 2015a;b).

Here we intend to further the understanding of the planetary burden caused by global
warming and the effect of the continued increase of GHG emissions from a new, a
rheological perspective. We focus on carbon ($CO_2$) emissions from fossil fuel burning and
land use between 1959 and 2015 (with the increase between 1850 and 1958 serving as
upstream emissions).[5] From the standpoint of a global observer, we see that as a consequence
of the increase, the $CO_2$ concentration in the atmosphere increases (rather quickly).
Concomitantly, the atmosphere warms (here combining the effect of tropospheric warming
and stratospheric cooling) and expands (by approximately 15–20 m in the troposphere per
decade since 1990), while part of the carbon is locked away (rather slowly) in land and
oceans, likewise under the influence of global warming (Global Carbon Project, 2019;
Lackner et al., 2011; Philipona et al., 2018; Steiner et al., 2011; Steiner et al., 2020).

It is not known how reversible and how much out of sync the latter process is in relation to
the former (Boucher et al., 2012; Dusza et al., 2020; Garbe et al., 2020; Schwinger and
Tjiputra, 2018; Smith, 2012). All we know is that the slower process remembers the influence
of the faster one which runs ahead. Here we ask three (nontrivial) questions: (1) Can this
global-scale memory—Earth's memory—be quantified? (2) Is Earth's memory a buffer


which is negligently exploited; and in the case that it is even a limited buffer, what is the
degree of exploitation? And (3) does Earth's memory allow its persistence (path dependency)
to be quantified? To the best of our knowledge, the answers to these questions are pending.
To get a grip on Earth's memory, we focus on the slow-to-fast temporal offset inherent in the
atmosphere–land/ocean system, while preferring an approach which is "as simple as possible
but no simpler"; i.e. here, which does not come at the cost of complexity. To this end, it is
sufficient to resolve subsystems as a whole and to perceive their physical reaction in response
to the increase in atmospheric $CO_2$ concentrations as a combined one (i.e., including effects
such as that of global warming). We refer to the subsystems' temporally disjunct reactions
hereafter as the expansion of the atmosphere by volume and the sequestration of carbon by
sinks. Under optimal conditions (referring to the long-term stability of the temporal offset),
the temporal-offset view even suggests that we can refrain from disentangling the exchange
of both thermal energy and carbon throughout the atmosphere–land/ocean system, as it is
done in climate-carbon models ranging from simple to complex (Flato et al., 2013; Harman
and Trudinger, 2014). The additional degree of simplicity will prove an advantage in
advancing our understanding of the temporal offset in terms of memory and persistence.
In view of the aforementioned questions, we chose a rheological stress-strain ($\sigma$-$\varepsilon$) model
(Roylance, 2001; TU Delft, 2021); here a Maxwell body (MB) consisting of an elastic
element (its constant, traditionally denoted $E$ [Young's modulus], is replaced by the
compression modulus $K$) and a damping (viscous) element (the damping constant is denoted
$D$), to capture the stress-strain behaviour of the global atmosphere–land/ocean system (Fig. 1)
and to simulate how humankind propelled that global-scale experiment historically.





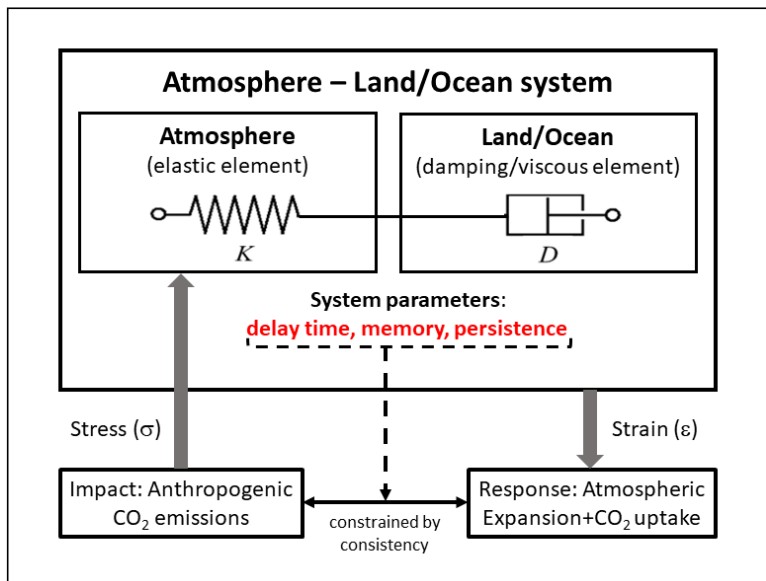

**Fig. 1:** Rheological model to capture the stress–strain behavior of the global atmosphere–

land/ocean system as a Maxwell body, consisting of elastic (atmosphere) and

damping/viscous (land/ocean) elements. The stress–strain behaviour is adjusted until

consistency is achieved (see text).

In practice, rheology is principally concerned with extending continuum mechanics to

characterise the flow of materials that exhibit a combination of elastic, viscous, and plastic

behaviour by properly combining elasticity and (Newtonian) fluid mechanics. Limits (e.g.,

viscosity limits) exist beyond which basic rheological models are recommended to be refined.

However, these limits are fluent, and basic rheological models also produce useful results

beyond these limits (Mezger, 2006; TU Delft, 2021).

Depending on whether the strain ($\varepsilon$) or the stress ($\sigma$) is known (in addition to the compression

and damping characteristics $K$ and $D$), the stress-strain equation describing a MB can be

applied in a stress-explicit form



$\sigma(t) = \sigma(0)\, exp\left(-\frac{K}{D}t\right) + K \int_0^t \dot{\varepsilon}(\tau)\, exp\left(\frac{K}{D}(\tau - t)\right) d\tau$    (1a)
or in a strain-explicit form
$\varepsilon(t) = \varepsilon(0) + \frac{1}{K}[\sigma(t) - \sigma(0)] + \frac{1}{D}\int_0^t \sigma(\tau)\, d\tau,$    (1b)
with $\sigma(0)$ and $\varepsilon(0)$ denoting initial conditions and a dot the derivative by time (Roylance,
2001; Bertram and Glüge, 2015).
For an observer it is the overall strain response of the atmosphere–land/ocean system
(expansion of the atmosphere by volume and uptake of $CO_2$ by sinks) that is unknown.
However, since atmospheric $CO_2$ concentrations have been observed to increase
exponentially (quasi continuously), the strain can be expected to be exponential or close to
exponential. In addition, we provide independent estimates of the likewise unknown
compression and damping characteristics of the MB. This a priori knowledge allows
equations (1a) and (1b) to be used stepwise in combination to narrow down our initial
estimate of the $K/D$ ratio, in particular. More accurate knowledge of this ratio is needed
when we go beyond textbook knowledge by distilling three parameters—delay time
(reflecting the temporal offset mentioned above), memory, and persistence—from the stress-
explicit equation. The three parameters depend, ceteris paribus, solely on the system's
characteristic $K/D$ ratio and allow deeper insights into that system. We see the atmosphere–
land/ocean system as being trapped progressively over time in terms of persistence. Given its
reduced ability to build up memory, we expect system failures globally well before 2050 if
the current trend in emissions is not reversed immediately and sustainably.
There exists a wide range of other approaches which aim at exploring memory and
persistence in Earth systems data, typically with the focus on individual Earth subsystems or
processes (e.g., atmospheric temperature or carbon dioxide emissions). So far, applied



approaches are mainly based on classical time-series and time-space analyses to uncover the
memory or causal patterns contained in observational data (Barros et al., 2016; Belbute and
Pereira, 2017; Caballero et al., 2002; Franzke, 2010; Lüdecke et al., 2013). However, these
approaches come with well-known limitations which can all be attributed, directly or
indirectly, to the issue of forecasting (more precisely, the conditions placed on the data to
enable forecasting) or are not based on physics (Aghabozorgi et al., 2015; Darlington, 1996;
Darlington and Hayes, 2016). By way of contrast, we do not forecast. We perpetuate long-
term historical conditions which, in turn, allows the delay time in the atmosphere–land/ocean
system to be expressed analytically in terms of memory and persistence.
Rheological approaches are common in Earth systems modeling as well. Typically, they are
applied to mimic the long(er)-term behaviour of Earth subsystems, e.g. its mantle viscosity
which is crucial for interpreting glacial uplift resulting from changes in planetary ice sheet
loads (Müller, 1986; Whitehouse et al. (2019); Yuen et al., 1986). Yet, to the best of our
knowledge, a rheological approach to unravel the memory-persistence behaviour of the
global atmosphere–land/ocean system in response to the long-lasting increase in atmospheric
$CO_2$ emissions had not been applied before.
We describe our rheological model (MB) approach in detail in Section 2, while we provide an
overview of the applied data and conversion factors in Section 3. In Section 4 we describe
how we derive first-order estimates of the main characteristics of the atmosphere–land/ocean
system (in terms of the MB's $K$ and $D$ characteristics) by using available knowledge.
Although uncertain, these estimates come useful in Section 5 where we apply the
aforementioned stress and strain explicit equations to quantify delay time, memory, and

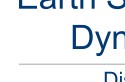
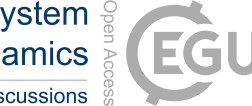

persistence of the atmosphere–land/ocean system. We conclude by taking account of our
main findings in Section 6.
**2.    Method**
We assume that we know the order of magnitude of both the $K/D$ ratio characteristic of the
atmosphere–land/ocean system and the rate of change in the strain $\varepsilon$ given by
$\dot{\varepsilon}(t) = \alpha \, exp(\alpha t)$ with the exponential growth factor $\alpha > 0$. These first-order estimates
permit equations (1a) and (1b) to be used stepwise in combination:
Equation (1a):  We vary both $K/D$ and $\alpha$ to reproduce the known stress $\sigma$ given by the $CO_2$

10                       emissions from fossil fuel burning (fairly well known) and land use (less known)

11                       (Global Carbon Project, 2019).

Equation (1b): We insert both the fine-tuned $K/D$ ratio and the known stress $\sigma$ to compute

13                       the strain $\varepsilon$ and check its derivative by time.

We consider this procedure a check of consistency, not a proof of concept.
Delay time, memory, and persistence are characteristic of the MB and are defined
independently of initial conditions. Thus, we rewrite equation (1a) for $\sigma(0) = 0$, which
results in
$\sigma(t) = \frac{D}{\beta} \dot{\varepsilon}(t) \left(1 - q_\beta^t\right)$                                    (2a)
(see Supplementary Information 1), where $\beta = 1 + \frac{D}{K}\alpha$ and $q_\beta^t = exp\left(-\frac{K}{D}\beta t\right)$. The term $\frac{D}{K\beta}$
represents a time characteristic of the MB under (here) exponential strain (i.e., of the MB that
responds to the stress acting upon it), whereas $\frac{D}{K}$ is the relaxation time of the MB (i.e., of the
MB that relaxes unhindered after the stress causing that strain has vanished, or that responds
to strain held constant over time; also known as the relaxation test (Bertram and Glüge,





2015). However, to ensure that exponents still come in units of 1 after we split them up, we
introduce the dimensionless time $n = \frac{t}{\Delta t}$ globally (which will be discretised in the sequel
when we refer to a temporal resolution of 1 year and set $\Delta t = 1y$), such that, for example,
$q^t = exp\left(-\frac{K}{D}\Delta t\right)^n.$
To understand the systemic nature of the MB, we explore here its stress dependence on
$q = exp\left(-\frac{K}{D}\Delta t\right)$, which contains the ratio of $K$ and $D$, the two characteristic parameters of
the MB, by way of derivation by $q$ (while $\alpha$ is held constant). To this end, we transform
equation (2a) further to
$\sigma_D(q,t) := \frac{1}{D}\sigma(t) = \frac{1}{D}\sigma(n) =: \sigma_D(q,n)$      (2b)
and execute $\frac{\partial}{\partial q}\sigma_D(q,n)$, the derivation by $q$ of the system's rate of change $\sigma_D$ (which is given
in units of y$^{-1}$). Doing so allows (what we call) delay time $T$ to be distilled (see
Supplementary Information 2). It is defined as
$T(q,n) := \frac{q_\beta}{S_n}\frac{\partial S_n}{\partial q_\beta} = -\frac{q_\beta^n}{1-q_\beta^n}n + \frac{q_\beta}{1-q_\beta},$      (3)
where $q_\beta = q_\alpha q$, $q_\alpha = exp(-\alpha\Delta t)$, and $S_n = S(q,n) = \frac{1-q_\beta^n}{1-q_\beta}$. The delay time behaves
asymptotically for increasing n and approaches $T_\infty = \lim_{n\to\infty}T = \frac{q_\beta}{1-q_\beta}$. We further define
$M := S(q,n)$      (4)
with $M_\infty := \frac{1}{1-q_\beta}$ and
$P := T(q,n)^{-1}$      (5)
with $P_\infty := \frac{1}{T_\infty} = \frac{1-q_\beta}{q_\beta}$ as the MB's characteristic memory and persistence, respectively. As is
commonly done, we keep the list of independent parameters minimal. (We only allow $K$ and
$D$ [i.e., $q$] in addition to $n$; see equations [2b] and [3]–[5], in particular.)





$T$ as given by equation (3) is not simply characteristic of the MB described by equation (2); it
can be shown to appear as delay time in the argument of any function dependent on current
and previous times, with a weighting decreasing exponentially backward in time (see
Supplementary Information 3). Equation (4) reflects the history the MB was exposed to
systemically prior to current time $n$ (during which $\alpha$ was constant; see Supplementary
Information 4). Equation (5) can be shortened to $T \cdot P = 1$. If we assume that $q$ can be
changed in retrospect at $n = 0$, this equation tells us that if $T$ —that is, $\Delta M$ per $\Delta q$ (or,
likewise, $\Delta M / M$ per $\Delta q / q$; see the first part of equation [3])—is small, $P$ is great because the
change in the system's characteristics (contained in $q$) hardly influences the MB's past, with
the consequence that the past exhibits a great path dependency, and vice versa.
An additional quantity to monitor is $ln(M \cdot P)$, which approaches $\lambda_\beta = \lambda \cdot \beta$ for increasing $n$
with $\lambda = \frac{K}{D} \Delta t$ the characteristic rate of change in the MB. The ratio $\lambda / ln(M \cdot P) \cdot$ allows
monitoring of how much the system's natural rate of change is exceeded as a consequence of
the continued increase in stress (see Supplementary Information 5).
**3.   Data and Conversion Factors**
A detailed overview of the carbon data and conversion factors used in this paper is given in
Supplementary Information 6. The data pertain to atmosphere, land, and oceans and are given
by source and time range and are also described briefly. The context within which they are
used is revealed in each of the following sections.
**4.   Independent Estimates of $D$ and $K$**





In this section we provide independent estimates of the damping and compression
characteristics of the atmosphere–land/ocean system, with $D_L$ and $D_O$ denoting the damping
constants assigned to land and oceans, respectively, and $K$ denoting the compression modulus
assigned to the atmosphere. We capture the characteristics' right order of magnitude
only—which can be done on physical grounds by evaluating the combined (net) strain
response of each subsystem on grounds of increasing $CO_2$ concentrations in the atmosphere.
These first-order estimates are adequate as they allow sufficient flexibility for Section 5,
where we narrow down our initial estimates by using equations (1a) and (1b) stepwise in
combination to achieve consistency.
**4.1   Estimating the Damping Constant $D_L$**
Increasing concentrations of $CO_2$ in the atmosphere trigger the uptake of carbon by the
terrestrial biosphere. The intricacies of this process, including potential (positive and
negative) feedback processes, are widely discussed (Dusza et al., 2020; Smith, 2012;
Heimann and Reichstein, 2008). The crucial question is how we have observed the process of
carbon uptake by the terrestrial biosphere taking place in the past. Compared to the reaction
of the atmosphere to global warming (an expansion of the atmosphere by volume), we
consider this process to be long(er) term in nature and perceive it as a Newton-like (damping)
element.
Biospheric carbon uptake is described by the biotic growth factor
$$\beta_b = \frac{\Delta NPP/NPP}{\Delta CO_2/CO_2}, \tag{6}$$
which is used to approximate the fractional increase in net primary production ($NPP$) per unit
increase in atmospheric $CO_2$ concentration (Amthor and Koch, 1996; Wullschleger et al.,
1995). Here we make use of the model-derived $NPP$ time series (1900–2016) provided by





O'Sullivan et al. (2019) to calculate $\beta_b$ (O'Sullivan et al., 2019). To understand the
uncertainty range underlying $\beta_b$, we use the photosynthetic beta factor
$\beta_{Ph} = CO_2 L = \left(\dfrac{dPh}{Ph}\right)\left(\dfrac{CO_2}{dCO_2}\right),$                   (7)
where $L$ is the so-called leaf-level factor denoting the relative leaf photosynthetic response to
a 1 ppmv change in the atmospheric concentration of $CO_2$, where
$L_1 \leq L = f(CO_2) \leq L_2,$                   (8)
and $Ph$ is the global photosynthetic carbon influx for 1959–2018. Equation (7) is similar to
equation (6). In equation (6) $\beta_b$ represents biomass production changes in response to $CO_2$
changes, whereas in equation (7) $\beta_{Ph}$ describes photosynthesis changes in response to $CO_2$
changes (Luo and Mooney, 1996).
$L$ can be shown to be independent of plant characteristics, light, and the nutrient environment
and to vary little by geographic location or canopy position. Thus, $L$ is virtually a constant
across ecosystems and a function of time-associated changes in atmospheric $CO_2$ only (Luo
and Mooney, 1996).
We use equation (7) to test whether $\beta_b$ falls in between the quantifiable photosynthetic limits
$L_1$ (photosynthesis limited by electron transport) and $L_2$ (photosynthesis limited by rubisco
activity). Fig. 2 shows the biotic growth factors from O'Sullivan et al. that consider changes
in $NPP$ due to the combined effect of $CO_2$ fertilisation, nitrogen deposition, climate change,
and carbon–nitrogen synergy ($\beta_{NPP\_comb}$) and due to $CO_2$ fertilisation ($\beta_{NPP\_CO2}$) only. For
1960–2016, $\beta_{NPP\_comb}$ falls in between $L_1$ and $L_2$, closer to $L_1$ than to $L_2$, whereas $\beta_{NPP\_CO2}$
falls even below the lower $L_1$ limit.

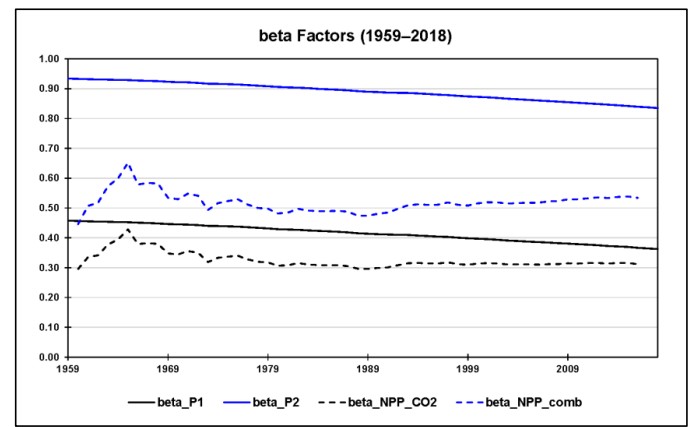

**Fig. 2:** Using the lower ($\beta_1$) and upper ($\beta_2$) limits of the photosynthetic beta factor to test the

range of the biotic growth factor ($\beta_b$) for 1960–2016. The biotic growth factor is

derived with the help of modelled net primary production ($NPP$) values provided by

$CO_2$ fertilisation, nitrogen deposition, climate change, and carbon–nitrogen synergy.

$\beta_{NPP\_CO2}$ refers to O'Sullivan et al. (2019),[35]who consider the change in $NPP$ due to

$CO_2$ fertilisation only, and $\beta_{NPP\_comb}$ refers to the change in $NPP$ due to the

combined effect.

Rewriting equation (7) in the form
$\dfrac{\Delta Ph_i}{Ph} = L_i \Delta CO_2 \quad (i = 1,2)$                                          (9)
with $Ph = 120 PgCy^{-1}$ indicating that the additional amount of annual relative
photosynthetic carbon influx, stimulated by a yearly increase in atmospheric $CO_2$
concentration, can be estimated by $L_i$, or the sequence of $L_i$ if $\Delta CO_2$ spans multiple years (see
Supplementary Information 7 and Supplementary Data 1). Plotting $\Delta Ph_i/Ph$ against time
allows lower and upper slopes (rates of strain)
$\dfrac{d}{dt}\left(\dfrac{\Delta Ph_1}{Ph}\right) \approx 0.0019 y^{-1}$ and $\dfrac{d}{dt}\left(\dfrac{\Delta Ph_2}{Ph}\right) = 0.0041 y^{-1}$              (10a,b)





to be derived for 1959–2018. A linear fit works well in either case. The cumulative increase
in atmospheric $CO_2$ concentration since 1959, $\Delta CO_2 = CO_2(t) - CO_2(1959)$, exhibits a
moderate exponential (close to linear) trend. Thus, plotting annual changes in $CO_2$,
normalised on the aforementioned rates of strain, versus time allows the remaining
(moderate) trends to be interpreted alternatively, namely, as average photosynthetic damping
constants with appropriate uncertainty given by half the maximal range (see Fig. 3 and
Supplementary Data 1)
$D_1 \approx (815 \pm 433)ppmv\,y = (83 \pm 44)Pa\,y = (2606 \pm 1383)10^6 Pa\,s$     (11a)
$D_2 \approx (378 \pm 201)ppmv\,y = (38 \pm 20)Pa\,y = (1207 \pm 641)10^6 Pa\,s$     (11b)

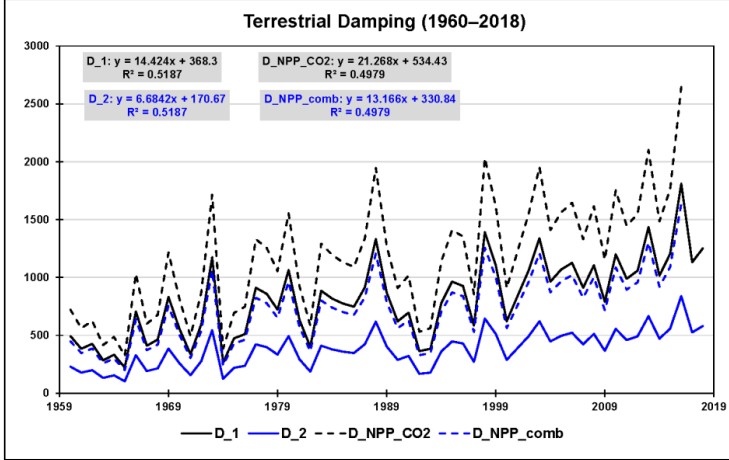

**Fig. 3:** Terrestrial carbon uptake perceived as damping (in ppmv y) based on the limits of leaf

photosynthesis (1960–2018: $D_1$) and $D_2$) and on model-derived changes in net

primary production ($NPP$; 1960–2016) due to both the combined effect of $CO_2$

fertilisation, nitrogen deposition, climate change, and carbon–nitrogen synergy

($D_{NPP\_comb}$) and $CO_2$ fertilisation only ($D_{NPP\_CO2}$). The linear trends of the four

damping series are shown at the top. These are used to interpret damping as constants

with appropriate uncertainty (given by half the maximal range).





Repeating the same procedure for 1959–2016 with O'Sullivan et al.'s model-derived $NPP$
values considering the change in $NPP$ due to $CO_2$ fertilisation as well as the total change in
$NPP$, we find
$\frac{d}{dt}\left(\frac{\Delta NPP}{NPP}\right)_{CO2} \approx 0.0013y^{-1}$ and $\frac{d}{dt}\left(\frac{\Delta NPP}{NPP}\right)_{comb} = 0.0021y^{-1}$       (12a,b)
(linear fits still work well); and consequently
$D_{CO2} \approx (1172 \pm 617)ppmvy = (119 \pm 62)Pay = (3746 \pm 1971)10^6 Pas.$     (13a)
$D_{comb} \approx (726 \pm 382)ppmvy = (74 \pm 39)Pay = (2319 \pm 1220)10^6 Pas.$     (13b)
As before, these estimates are closer to the lower leaf-level factor (higher photosynthetic $D$)
than to the higher leaf-level factor (lower photosynthetic $D$; Fig. 3).
Here we interpret O'Sullivan et al.'s Earth systems model as a typical one, which means that
the $NPP$ changes it produces are common. We therefore (and sufficient for our purposes)
choose the damping constant $D_1$ as a good estimator of the total change in $NPP$ of the
terrestrial biosphere since 1960. Hence
$D_L \approx (815 \pm 433)ppmvy = (83 \pm 44)Pay = (2606 \pm 1383)10^6 Pas.$     (14)
$D_L$ is on the order of viscosity indicated for bitumen/asphalt (Mezger, 2006).
**4.2  Estimating the Damping Constant $D_O$**
Increasing concentrations of $CO_2$ in the atmosphere trigger the uptake of carbon by the
oceans (National Oceanic and Atmospheric Administration, 2017). Like the uptake of carbon
by the terrestrial biosphere, we consider this process to behave like a Newton (damping)
element in our MB because of the irreversibility (due to hysteresis) on the shorter time scale
we are interested in (Schwinger and Tjiputra, 2018).





The Revelle (buffer) factor ($R$) quantifies how much atmospheric $CO_2$ can be absorbed by
homogeneous reaction with seawater. $R$ is defined as the fractional change in $CO_2$ relative to
the fractional change in dissolved inorganic carbon ($DIC$):
$R = \frac{\Delta pCO_2 / pCO_2}{\Delta DIC / DIC}$.  (15)
(Here, in contrast to before, atmospheric $CO_2$ is referred to in units of µatm and therefore
indicated by $pCO_2$.) An $R$ value of 10 indicates that a 10% change in atmospheric $CO_2$ is
required to produce a 1% change in the total $CO_2$ content of seawater (Bates et al. 2014;
Egleston et al., 2010; Emerson and Hedges, 2008).
$DIC$ and $R$ have been observed at seven ocean carbon time-series sites for periods from 15 to
30 years (between 1983 and 2012) to change slowly and linearly with time (Bates et al.

13  2014):

$\frac{\Delta DIC}{\Delta t} \approx [0.8; 1.9] \mu mol\, kg^{-1} y^{-1}$  (16)
$\frac{\Delta R}{\Delta t} \approx [0.01; 0.03] y^{-1}$  (17)
(see also Supplementary Data 2). Here it is sufficient to proceed with spatiotemporal
averages. As before, the cumulative increase in atmospheric $CO_2$ concentration since 1983,
$\Delta pCO_2 = pCO_2(t) - pCO_2(1983)$, exhibits a moderate exponential (close to linear) trend.
Thus, plotting annual changes in $pCO_2$, normalised on the rates of strain $\frac{(\Delta DIC / DIC)}{\Delta t}$, versus
time allows the remaining (moderate) trend to be interpreted alternatively, namely, as an
average oceanic damping constant with appropriate uncertainty given by half the maximal
range (see Fig. 4 and Supplementary Data 2):
$D_O \approx (3005 \pm 588) ppmv\, y = (304 \pm 60) Pa\, y = (9602 \pm 1877) 10^6 Pa\, s$.  (18)



$D_O$ is on the order of viscosity indicated for bitumen/asphalt, yet approximately 3.7 times
greater than $D_L$.

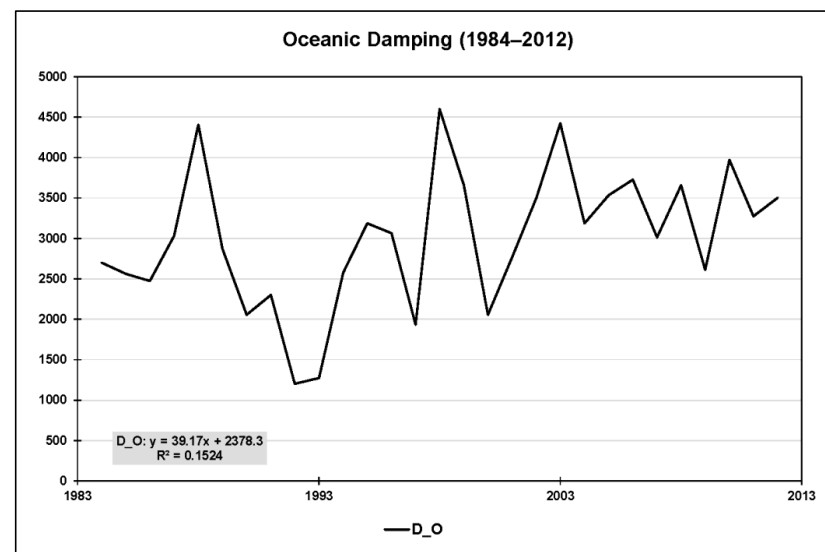

**Fig. 4:** Oceanic carbon uptake perceived as damping (in ppmv y) based on observations at
seven ocean carbon time-series sites for periods from 15 to 30 years (between 1983
and 2012). The linear trend in oceanic damping, shown at the bottom, is used to
interpret damping as a constant with appropriate uncertainty (given by half the
maximal range).
**4.3    Estimating the Compression Modulus $K$**
The long-lasting increase in GHG emissions has caused the $CO_2$ concentration in the
atmosphere to increase and the atmosphere as a whole to warm (with tropospheric warming
outstripping stratospheric cooling) and to expand (in the troposphere by approximately
15–20 m per decade since 1990) (Global Carbon Project, 2019; Lackner et al., 2011;
Philipona et al., 2018; Steiner et al., 2011; Steiner et al., 2020). Our whole-subsystem (net-



warning) view does not invalidate the known facts that $CO_2$ in the atmosphere is well-mixed
(except for very low altitudes where deviations from uniform $CO_2$ concentrations are caused
by the dynamics of carbon sources and sinks) and that the volume percentage of $CO_2$ in the
atmosphere stays almost constant up to high altitudes Abshire et al., 2010; Emmert et al.,

5    2012).

Compared to the slow uptake of carbon by land and oceans, we assume the atmosphere to be
represented well by a Hooke element in the MB and this to serve as a (sufficiently stable)
surrogate physical descriptor for the reaction of the atmosphere as a whole (Sakazaki and
Hamilton, 2020). However, in the case of a gas, Young's modulus $E$ must be replaced by the
compression modulus $K$, the reciprocal of which is compressibility $\kappa$. Both $K$ and $\kappa$ scale
with altitude which we get to grips with in the following. Compressibility is defined by
$\kappa = \frac{1}{K} = -\frac{1}{V}\frac{dV}{dp}$                (19)
($\kappa > 0$) (OpenStax, 2020). Depending on whether the compression happens under isothermal
or adiabatic conditions, the compressibility is distinguished accordingly. It is defined by
$\kappa_{it} = \frac{1}{p}$                (20a)
in the isothermal case and
$\kappa_{ad} = \frac{1}{\gamma p}$                (20b)
in the dry adiabatic case, where $\gamma$ is the isentropic coefficient of expansion. Its value is 1.403
for dry air (1.310 for $CO_2$) under standard temperature (273.15 K) and pressure (1 atm;
101.325 kPa) (Wark, 1983). We consider a carbon-enriched atmosphere also as air.
However, the observed expansion of the troposphere happens neither isothermally nor dry-
adiabatically but polytropically. Moreover, our ignorance of the exact value of $\kappa$ is



overshadowed by the uncertainty in altitude—or top of the atmosphere (TOA)—which we
need as a reference for $\kappa$ (thus $K$). As a matter of fact, there exists considerable confusion as
to which altitude the TOA refers in climate models (CarbonBrief, 2018; NASA Earth
Observatory, 2006).
To advance, we make reference to the (dry adiabatic) standard atmosphere, which assigns a
temperature gradient of –6.5°C/1000 m up to the tropopause at 11 km, a constant value of –
56.5°C (216.65 K) above 11 km and up to 20 km, and other gradients and constant values
above 20 km (Cavcar, 2000; Mohanakumar, 2008). Guided by the distribution of atmospheric
mass by altitude, we choose the stratopause as our TOA (at about 48 km altitude and 1 hPa),
with uncertainty ranging from mid-to-higher stratosphere (at about 43 km altitude and 1.9
hPa) to mid-mesosphere (at about 65 km altitude and 0.1 hPa) (Digital Dutch, 1999;
International Organization for Standardization, 1975; Mohanakumar, 2008; Zellner, 2011).
We assign the resulting uncertainty of 90% in relative terms to
$K = (1 \pm 0.9)hPa = (100 \pm 90)Pa,$          (21)
which we consider sufficiently large to compensate for the unknown isentropic coefficient in
the first place; that is, $[K_{ad,min}; K_{ad,max}] \in [K_{it,min}; K_{ad,max}] \in [K_{min}; K_{max}]$. For
comparison, $K_{ad}$ would range from 400 to 412 hPa were the TOA allocated within the
troposphere (exhibiting, the reference used here, an expansion of 20 m; see Supplementary
Information 8).
**5.**    **Main Findings** (1837 words)
Equation (1a) (or [2a], respectively) and equation (1b) are used stepwise in combination to
conduct three sets of stress-strain experiments including sensitivity experiments (SEs):
**A.** for the period 1959–2015 assuming zero stress and strain in 1959,





**B.** for the period 1959–2015 assuming zero stress and strain in 1900, and
**C.** for the period 1959–2015 assuming zero stress and strain in 1850.
The logic of the experiments is determined by both the availability of data (see
Supplementary Information 6) and the increasing complementarity from A to C (see below).
The basic procedure is always the same: We insert into equation (1a) our first-order estimates
of $D_L \approx (83 \pm 44) Pay$; $D_O \approx (304 \pm 60) Pay$, that is, $D = D_L + D_O \approx (387 \pm 74) Pay$;
and $K \approx (100 \pm 90) Pa$. At the same time, we use the growth factor $\alpha_{ppm} = 0.0043 y^{-1}$,
which reflects the exponential increase in the $CO_2$ concentration in the atmosphere between
1959 and 2018 (see Supplementary Data 1) as our first-order estimate for $\alpha$ in
$\dot{\varepsilon} = \alpha \, exp(\alpha t)$, the rate of change in strain $\varepsilon$. We apply equation (1a) by varying both $K/D$
and $\alpha$ to reproduce the known stress $\sigma$ on the left, given by the $CO_2$ emissions from fossil
fuel burning and land use. To restrict the number of variation parameters to two, we let $K$ and
$D$ deviate from their respective mean values equally in relative terms (i.e., we assume that our
first-order estimates exhibit equal inaccuracy in relative terms) and express $\alpha$ as a multiple of
$\alpha_{ppm}$. This is easily possible with the introduction of suitable factors (see Supplementary
Data 3) that allow $\sigma$ to be reproduced quickly and with sufficient accuracy. The main reason
this works well is that the two factors pull the two exponential functions on the right side of
equation (2a)—$\dot{\varepsilon}(t)$ and $\left(1 - q_{\beta}^{t}\right)$, which determine the quality of the fit—in different
directions.
**To A**
This is our set of reference experiments, all for the period 1959–2015. This set comprises
**A.1)** a stress-explicit experiment, **A.2)** three strain-explicit experiments, and **A.3)** SEs





expanding the strain-explicit experiments. The parameters $\alpha$, $\lambda$, and $\lambda_\beta$ are reported in $\mathrm{y}^{-1}$, as
is commonly done.
**To A.1:** In this experiment we vary the ratio $K/D$ ($\lambda$ in Table 1) and $\alpha$ to reproduce the
monitored stress $\sigma(t)$ on the left side of equation (2a) (see Supplementary Data 3). This
tuning process (hereafter referred to as "Case 0") allows us to test whether $K$ and $D$, in
particular, stay within their estimated limits, namely, $K \in [10; 190]Pa$ and
$D \in [313; 461]Pa\,y$ or, equivalently, $\lambda \in [0.0217; 0.6078]y^{-1}$. Column "Case 0" in Table 1
indicates that this case is practically identical to choosing $\lambda = (10/461)y^{-1} = 0.0217y^{-1}$,
the smallest ratio $K/D$ deemed possible. For Case 0 we find $K = 9.9Pa$ and $D = 461.5Pa\,y$
(thus, $\lambda = K/D = 0.0214y^{-1}$) and, concomitantly, $\alpha = 0.0247y^{-1}$ (thus,
$\lambda_\beta = (K/D)\beta = (K/D) + \alpha = 0.0461y^{-1}$).
**Table 1:** Overview of parameters in experiments A.1–A.3.

| Parameter | | Case 0 | Case 1 | Case 12 | Case 13 | Case 2 | Case 21 | Case 23 | Case 3 | Case 31 | Case 32 |
|---|---|---|---|---|---|---|---|---|---|---|---|
| | | stress explicit | strain explicit | sensitivity experiments Case 1 | | strain explicit | sensitivity experiments Case 2 | | strain explicit | sensitivity experiments Case 3 | |
| **K** | Pa | 9.9 | 10 | 10 | 10 | 100 | 100 | 100 | 190 | 190 | 190 |
| **D** | Pa y | 461.5 | 461 | 461 | 461 | 387 | 387 | 387 | 313 | 313 | 313 |
| $\lambda$ [a,b] | $\mathrm{y}^{-1}$ | 0.0214 | 0.0217 | 0.0217 | 0.0217 | 0.2584 | 0.2584 | 0.2584 | 0.6078 | 0.6078 | 0.6078 |
| $\lambda^{-1}$ | y | 46.8 | 46.1 | 46.1 | 46.1 | 3.87 | 3.87 | 3.87 | 1.65 | 1.65 | 1.65 |
| $\alpha$ [a] | $\mathrm{y}^{-1}$ | 0.0247 | 0.0248 | 0.0158 | 0.0174 | 0.0158 | 0.0248 | 0.0174 | 0.0174 | 0.0248 | 0.0158 |
| $\beta$ | 1 | 2.158 | 2.144 | 1.729 | 1.803 | 1.061 | 1.096 | 1.067 | 1.029 | 1.041 | 1.026 |
| $\lambda_\beta$ [a] | $\mathrm{y}^{-1}$ | 0.0461 | 0.0465 | 0.0375 | 0.0391 | 0.2742 | 0.2832 | 0.2758 | 0.6252 | 0.6236 | 0.6236 |
| $\lambda_\beta^{-1}$ | y | 21.7 | 21.5 | 26.7 | 25.6 | 3.65 | 3.53 | 3.63 | 1.60 | 1.58 | 1.60 |
| $q_\beta$ | 1 | 0.9549 | 0.9546 | 0.9632 | 0.9617 | 0.7602 | 0.7534 | 0.7590 | 0.5351 | 0.5312 | 0.5360 |
| $T_\infty$ | 1 | 21.19 | 21.02 | 26.19 | 25.10 | 3.17 | 3.05 | 3.15 | 1.15 | 1.13 | 1.16 |
| $M_\infty = T_\infty/q_\beta$ | 1 | 22.19 | 22.02 | 27.19 | 26.10 | 4.17 | 4.05 | 4.15 | 2.15 | 2.13 | 2.16 |
| $P_\infty = 1/T_\infty$ | 1 | 0.0472 | 0.0476 | 0.0382 | 0.0398 | 0.3155 | 0.3274 | 0.3176 | 0.8686 | 0.8825 | 0.8657 |
| $\lambda/\lambda_\beta = 1/\beta$ | % | 46.3 | 46.6 | 57.8 | 55.5 | 94.2 | 91.2 | 93.7 | 97.2 | 96.1 | 97.5 |
| **n** at $T/T_\infty=0.5$ | 1 | --- | 28 | 34 | 33 | 5 | 5 | 5 | 3 | 3 | 3 |

| | | | | | | | | | | | |
|---|---|---|---|---|---|---|---|---|---|---|---|
| $\lambda$ / LN(M·P) | % | --- | 5 | 5 | 5 | 36 | 36 | 36 | 54 | 53 | 54 |
| n at M/M$_\infty$=0.5 | 1 | --- | 15 | 19 | 18 | 3 | 2 | 3 | 1 | 1 | 1 |
| $\lambda$/ln(M·P) | % | --- | 4 | 4 | 4 | 22 | 21 | 22 | n.a. | n.a. | n.a. |
| n at T/T$_\infty$=0.95 | 1 | --- | 98 | 121 | 116 | 17 | 17 | 17 | 8 | 8 | 8 |
| $\lambda$ / LN(M·P) | % | --- | 25 | 28 | 27 | 82 | 79 | 81 | 91 | 90 | 91 |
| n at M/M$_\infty$=0.95 | 1 | --- | 64 | 80 | 77 | 11 | 11 | 11 | 5 | 5 | 5 |
| $\lambda$ / LN(M·P) | % | --- | 13 | 13 | 13 | 61 | 60 | 61 | 74 | 74 | 74 |

[a] Given in $y^{-1}$.
[b] Derived for $K$ and $D$ deviating from their respective mean values equally in relative terms.

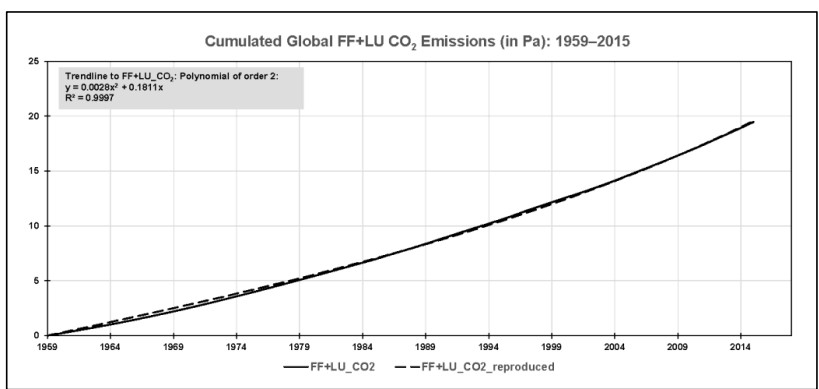

**Fig. 5:** Case 0: $K/D$ and $\alpha$ on the right side of equation (2a) are tuned to reproduce the stress
$\sigma(t)$ on the left side of that equation, given by the monitored (but cumulated) $CO_2$
emissions from fossil fuel burning and land use activities (in Pa).
Fig. 5 reflects the result of the tuning process graphically. It shows how well the monitored
stress, given by the cumulated $CO_2$ emissions from fossil fuel burning and land use activities
since 1959, can be reproduced by equation (2a). The quality of the tuning is observed by
summing the squares of differences between monitored and reproduced stress from 1959 to
2015 using the SUMXMY2 command in Excel. (We stopped the tuning process with the sum





at about 1.400 Pa$^2$, when changes in $K$ and $D$ became negligible, resulting in a correlation
coefficient of 0.9998; see Supplementary Data 3.)
Fig. 5 also shows the parameters needed to describe the monitored stress by a second-order
polynomial regression (see the grey box in the upper left corner of the figure). We have not
yet used this regression but will do so in the strain-explicit experiments described next.
**To A.2:** We use equation (1b) with $\sigma(0) = \varepsilon(0) = 0$ and $\sigma(t) = 0.0028t^2 + 0.1811t$, the
second-order polynomial regression of the monitored stress (cf. Fig. 5), to conduct three
experiments (hereafter referred to as "Cases 1–3") to explore the spread in the strain $\varepsilon$. To
this end, we let the ratio $K/D$ vary from minimum (Case 1) to mean (Case 2) to maximum
(Case 3; see Table 1 and Supplementary Data 4) irrespective of the outcome of the Case 0
experiment, which suggests that compared to Cases 2 and 3, Case 1 ($K$ minimal: the
atmosphere is rather compressible, $D$ maximal: the land and oceans are rather viscous)
appears to be more in conformity with reality than Cases 2 and 3.

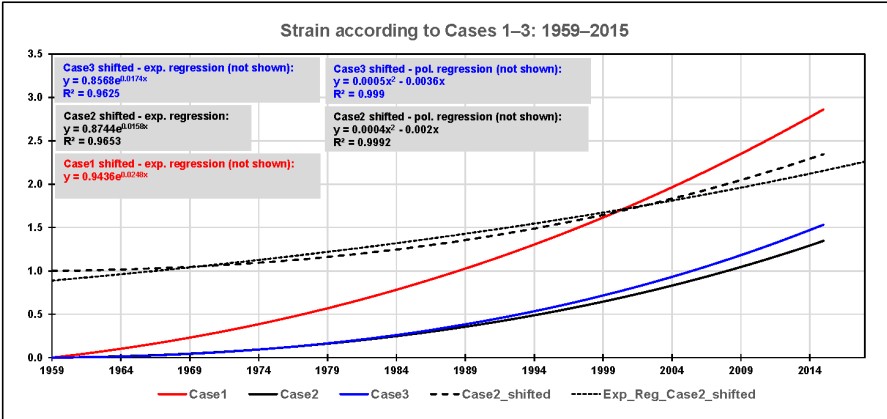

**Fig. 6:** Cases 1–3: The ratio $K/D$ is varied from minimum (Case 1: solid red) to mean (Case

19        2: solid black) to maximum (Case 3: solid blue) to explore the spread in the strain $\varepsilon$





(in units of 1) on the left side of equation (1b), while the monitored stress is described
by a second-order polynomial (see the text). These strain responses have to be shifted
upward (so that they pass through 1 in 1959) to derive their rates of change, if
described by an exponential regression (here only demonstrated for Case 2). As is
already illustrated in Case 0, the exponential regression in Case 1 is excellent (see the
text), whereas second-order polynomial regressions provide better fits in Cases 2 and
3 (see the boxes in the figure; the polynomial regressions are not shown).
Fig. 6 reflects these experiments graphically. It shows that the range of strain responses is
encompassed by Case 1 ($K/D = (10/461)y^{-1}$) and Case 2 ($K/D = (100/387)y^{-1}$), not
by Case 1 and Case 3 ($K/D = (190/313)y^{-1}$)—the solid blue line (Case 3) falls in between
the solid red (Case 1) and solid black (Case 2) lines—resulting from how $K$ and $D$ dominate
the individual parts of equation (1b). These strain responses have to be shifted upward (so
that they pass through 1 in 1959) to describe them by an exponential regression and to derive
their rates of change. The exponential fit is excellent only in Case 1, as already illustrated in
Case 0 (Case 0: $\lambda = 0.0214y^{-1}$, Case 1: $\lambda = 0.0217y^{-1}$), but inferior to the polynomial
regressions, here of the second order, in Cases 2 and 3. However, a second-order polynomial
approach to the strain has to be discarded because the stress derived with the help of equation
(1a) would exhibit a linear behaviour with increasing time and not be a polynomial of the
second order as in Fig. 6 (see Supplementary Information 9).



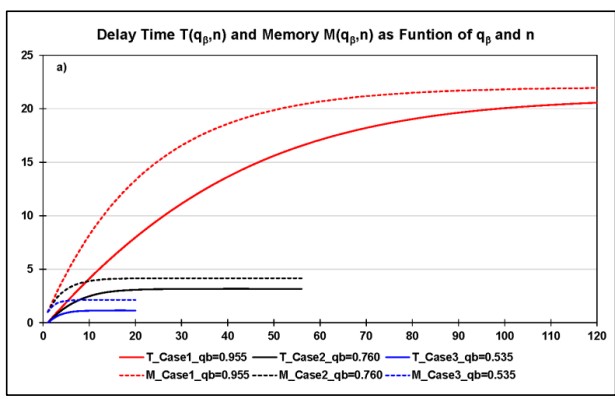

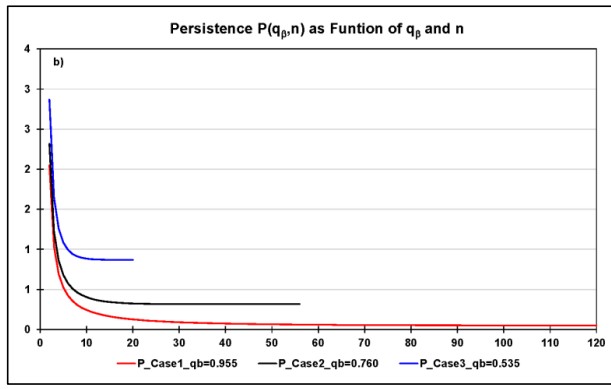

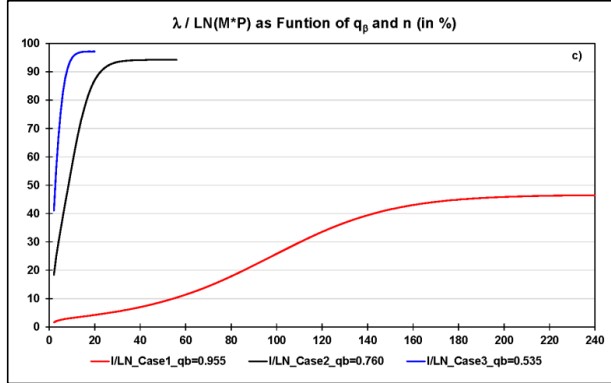

4    **Fig. 7:** Cases 1–3: **a)** delay time $T$ and memory $M$ (in units of 1), **b)** persistence $P$ (in units of

5        1), and **c)** the ratio $\lambda/ln(M \cdot P)$ (in %); all are versus time (in units of 1).




In this regard we note that a more targeted way forward would be to use a piecemeal
approach. This approach requires the data series to be sliced into shorter time intervals,
during which an exponential fit for the strain (which we assume to hold in principle in
deriving equation [2a] here) is sufficiently appropriate. Fortunately, as the SEs in A.3
indicate, we can hazard the consequences of using suboptimal growth factors resulting from
suboptimal exponential regressions for the strain.
Equations (3) to (5) are used to determine delay time $T$, memory $M$, and persistence $P$ (in
units of 1) for Cases 1–3 as well as their characteristic limiting values $T_\infty$, $M_\infty$, and $P_\infty$ (see
Table 1 and Supplementary Data 5 to 8). Fig. 7a and 7b reflect the behaviour of $T$, $M$, and $P$
over time (in units of 1). For a better overview, Table 1 lists the times when these parameters
exceed 50% or 95%, respectively, of their limiting values (without indicating whether these
levels go hand in hand with, e.g., global-scale ecosystem changes of equal magnitude). In the
table we also specify the ratio $\lambda/ln(M \cdot P)$ for each of these times (see also Fig. 7c). The
ratio approaches $\lambda/\lambda_\beta$ for $n \rightarrow \infty$ and indicates (as a percentage) how much smaller the
system's natural rate of change in the numerator turns out compared to the system's rate of
change in the denominator under the continued increase in stress. As is illustrated, in
particular, by Case 1 in the figure, the ratio does not increase at a constant pace as n
increases, which shows the nonlinear strain response of the atmosphere–land/ocean system.
**To A.3:** Three sets of SEs serve to assess the influence of the exponential growth factor on
the strain-explicit experiments described above:
**SE1:**    $\alpha_1 = 0.0248y^{-1}$ as in Case 1 (cf. Fig. 6) is also used in Cases 2 and 3 (hereafter

24        referred to as "Cases 21 and 31").





**SE2:**    $\alpha_2 = 0.0158 y^{-1}$ as in Case 2 (cf. Fig. 6) is also used in Cases 1 and 3 (hereafter
referred to as "Cases 12 and 32").
**SE3:**    $\alpha_3 = 0.0174 y^{-1}$ as in Case 3 (cf. Fig. 6) is also used in Cases 1 and 2 (hereafter
referred to as "Cases 13 and 23").
Table 1 shows that the influence of a change in the exponential growth factor is small vis-à-
vis the dominating influence of $K$ and $D$ and the quality in the estimates of $T$, $M$, and $P$. For
instance, the dimensionless time $n$ at $M/M_\infty = 0.5$ ranges from 15 to 19 in Case 1 and
Case 1–related experiments (small persistency) and from 2 to 3 in Case 2 and Case 2–related
experiments (great persistency); in Case 3 and Case 3–related experiments, it does not exhibit
a range at all ($n \approx 1$; very great persistency). These ranges for $n$ tell us how long it takes to
build up 50% of the memory with time running as of $n = 0$ (1959).
**Table 2:**    Cases 1–3 and related experiments: Build-up of memory (%) as of $n = 0$ (1959).

| Time | | Increase in memory as of n=0 (1959) | | |
|---|---|---|---|---|
| | | Cases 1, 12, 13 | Cases 2, 21, 23 | Cases 3, 31, 32 |
| y | 1 | % | % | % |
| 1959 [a] | 0 | 0.0 | 0.0 | 0.0 |
| 1964 | 5 | 17–21 | 75–76 | 96 |
| 1970 | 11 | 34–40 | 95–96 | 100 |
| 2015 | 56 | 88 – 93 | 100 | --- |

[a] Start year: $\sigma_0 = \varepsilon_0 = 0$.
Alternatively, we can ask how much memory has been build up until a given year. Table 2
tells us that after 56 years (i.e., in 2015) memory is still building up only in Case 1 and Case
1–related experiments, which means that the system still responds in its own characteristic
way (as a result of a small $K$ and a great $D$) to the continuously increasing stress; this is not
so in Cases 2 and 3 (and related experiments). In the latter two cases today's uptake of carbon
by land and oceans happens de facto outside the system's natural regime and solely in





response to the sheer, continuously increasing stress imposed on it, whereas in Case 1 and
Case 1–related experiments the limits of the natural regime are not yet reached. This
interpretation of Cases 1–3 (and related experiments) does not depend on how much carbon
the system already took up before 1959, because $M$ is additive and the current $M/M_\infty$ value
considers $M/M_\infty$ to be achieved historically (e.g., during the previous time interval) by way
of adjusting initial conditions.
Finally, it is important to note that it is prudent to expect that natural elements (like land and
oceans) will not continue to maintain their damping capacity—or their capacity to embark on
a, most likely, hysteretic downward path in the case of a sustained decrease in emissions—
even well before they reach the limits of their natural regimes. They may simply collapse
globally.
**To B and C**
We report on the sets of stress-strain experiments B and C in combination. They can be
understood as a repetition of the 1959–2015 Case 0 experiment (see A.1) but with the
difference that now upstream emissions as of 1900 (B) or 1850 (C), respectively, are
considered. This allows initial conditions for 1959 other than zero, as in the Case 0
experiment, to be taken into account (see Supplementary Information 10 and Supplementary
Data 9 to 16):
Case 0:   1959–2015
B:        1900–1958 (upstream emissions), 1959–2015
C:        1850–1958 (upstream emissions), 1959–2015



The experiments can be ordered consecutively in terms of time with the three 1959–2015
periods comprising a min–max interval to facilitate the drawing of a number of robust results
in spite of the uncertainty underlying these stress-strain experiments (see Supplementary
Information 10). Between 1850 and 1959–2015 (i) the compression modulus $K$ increased
from ~2 to 10–13 Pa (the atmosphere became less compressible) while (ii) the damping
constant $D$ decreased from ~468 to 459–462 Pa y (the land and oceans became less viscous),
with the consequence that (iii) the ratio $\lambda = K/D$ increased from ~0.004–0.005 $y^{-1}$ to 0.021–
0.028 $y^{-1}$ (i.e., by a factor of 4–6). Likewise, (iv) delay time $T_\infty$ decreased (hence persistence
$P_\infty$ increased) from ~51 (~0.02) to 18–21 (0.047–0.055) while (v) memory $M_\infty$ decreased
from ~52 to 19–22 on the dimensionless time scale.
**6.  Account of the Findings**
Our Case 0 experiment (see A.1) in combination with stress-strain experiments B and C
described above allows us to draw some precautionary conclusions. The values of the Case 0
parameters $T_\infty$ and $M_\infty$, in particular, are at the upper end of the respective 1959–2015 min–
max intervals (see Supplementary Information 10). That is, the respective characteristic ratios
$T/T_\infty$ and $M/M_\infty$ reach specified levels (e.g., 0.5 or 0.95; see Fig. 7a) slightly sooner than
when $T_\infty$ and $M_\infty$ take on values at the lower end of the 1959–2015 min–max intervals.
Given that Case 0 is well represented by Case 1 (see A.2), we can use the parameter values of
the latter. According to column "Case 1" in Table 1, $M/M_\infty$ and $T/T_\infty$ reached their 0.5 levels
after about 15 and 28 year-equivalent units on the dimensionless time scale (which was in
1974 and 1987), whereas they will reach their 0.95 levels after about 64 and 98 year-
equivalent units (which will be in 2023 and 2057) if the exponential growth factor $\alpha$ remains
unchanged in the future. However, the increase in $P_\infty$, here by a factor of 2–3, indicates that





the atmosphere–land/ocean system is progressively trapped in terms of persistence, which
means that it will become progressively more difficult to strain-relax the entire system (i.e.,
the atmosphere including land and oceans). A mere 1-year decrease of a few percentage
points in emissions, as reported recently for 2020, will have virtually no impact Global
Carbon Project, 2020).
We understand, in particular, the ability of a system to build up memory effectively as its
ability to respond still in its own characteristic way (i.e., within its natural regime; see A.3).
Therefore, it appears precautionary to prefer memory over delay time in avoiding potential
system failures globally in the future. These we expect to happen well before 2050 if the
current trend in emissions is not reversed immediately and sustainably.
We consider this statement robust given both the uncertainties we dealt with in the course of
our evaluation and the restriction of our variation parameters to two. One of the two variation
parameters ($\lambda$) presupposes knowing $K$ and $D$ with equal inaccuracy in relative terms. The
introduction of this parameter, in particular, offers a great applicational benefit, but no serious
restriction given that, while $\alpha$ is held constant, it is the $K/D$ ratio that counts and whose
ultimate value is controlled by consistency—which comes in as a powerful rectifier. As a
matter of fact, fulfilling consistency results in a $K/D$ ratio that ranges close to the lower
uncertainty boundary which we deem adequate based on our preceding assessment. That is, a
smaller $K$: the atmosphere is more compressible than previously thought; and a greater $D$:
land and oceans are more viscous than previously thought. However, the overall effect of the
continued release of GHG emissions since 1850 on the $K/D$ ratio is unambiguous—the ratio
increased by a factor 4–6 ($K$ increased: the atmosphere became less compressible; $D$





decreased: land and oceans became less viscous), resulting in the aforementioned changes in
delay time, memory, and persistence.
The latter two Earth system characteristics can be summarized in lieu of the questions posed
in the beginning: Earth's memory is a limited buffer, approximately 60% of which
humankind had already exploited prior to 1959; while its persistence (path dependency)
increases by approximately a factor 2–3 if the release of emissions globally continues as
before.



## Acknowledgements

Funding was provided by the authors' home institutions. Additional funding to facilitate

collaboration between the Lviv Polytechnic National University and IIASA was provided by

the bilateral Agreement on Scientific and Technological Co-operation between the Cabinet of

Ministers of Ukraine and the Government of the Republic of Austria (S&T Cooperation

Project 10/2019; https://oead.at/en/ and www.mon.gov.ua/). Net primary production, land-use

change emission, and atmospheric expansion data were kindly provided personally by

Michael O'Sullivan (University of Exeter), Julia Pongratz (Ludwig Maximilian University of

Munich), and Andrea K. Steiner (Wegener Center for Climate and Global Change, Graz).

## Data Availability

Supplementary Material (Supplementary Information and Supplementary Data):

https://doi.org/10.22022/em/06-2021.123

## Author Contributions

M.J. set up the physical model of the atmosphere–land/ocean system; derived its delay time,

memory, and persistence; and provided the initial estimates of its compression and damping

characteristics. R. B. contributed to the physical and mathematical improvement of the

method and the physical consistency of results. I. R. and P. Z. contributed to the inspection of

mathematical relations globally and their generalizations. P.Z. contributed to the

strengthening of the method by evaluating alternative memory concepts known in

mathematics.



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
