# Peer review of "Title: Quantifying memory and persistence in the atmosphere–land/ocean carbon system"

_Earth System Dynamics, 2021_

## Referee Comment (RC1)

The manuscript „Quantifying memory and persistence in the atmosphere–land/ocean carbon system" is really unusual and sticks out of usual approaches to model carbon cycle, and this is one of the reasons why I agreed to review it. In particular, I was curious how do the authors define the memory and persistence. However, as the field of climate and carbon cycle dynamics is well established and presented by a range of models from zero-dimensional to fully-fledged Earth System models, the authors have to link their definitions and approaches to the ones existing in the field. For example, what does resilience mean in terms of remaining carbon budget until climate goal? What are stress and strain of the Earth system, and how can we measure them? What does it mean that the memory is exploited?

The chosen model suggests the Earth System is under stress from high CO2. Rates of global warming and CO2 growth are indeed unprecedented, and ecosystems are in stress to adapt. However, high level of warming was seen in the Earth history many times and the term "stress" is simply not that easy to interpret.  I also anticipated that the model could be more universal and be applied to the other periods in the Earth periods such as glacial cycles.

I also expected that the model could explain fundamental features of the carbon system, for example, why the combined anthropogenic carbon uptake by land and ocean today is about 60%, and not 40% or 80%. Unfortunately, the model parameters are tuned to reproduce observations, assuming that pre-industrial state is without stress. Actually, most of the time during the last million years the Earth System was in cold conditions under low CO2 concentrations. If one uses the Last glacial maximum ca. 20 thousand years ago as a reference, would the pre-industrial state be a stress for the Earth System?

In summary, using the new approach such as rheological model is useful in case the method brings something new comparing to the current set of models used for policy applications. However, I just do not see what new can we learn from this approach. It looks more as reformulation of the problem in new variables which are non-measurable and have little content to be interpreted by policy makers, such as remaining carbon budget. I think the approach is interesting, so I do not recommend to reject the manuscript, but without substantial rewriting and interpreting terms in measurable quantities this study will miss the auditorium.

Specific comments

P.2, l.6: "**the CO2 concentration in the atmosphere increases (rather quickly).** Concomitantly, the atmosphere warms and expands, while **part of the carbon is locked away (rather slowly) in land and oceans**, likewise under the influence of global warming." This statement is in disagreement with data. Over last decades, airborne fraction of CO2 emissions is rather constant, see for example Chapter 5 in IPCC AR6 WGI (Canadell et al., 2021). That means that CO2 in atmosphere is increasing as fast as sinks on land or in the ocean.

l.11: "It is not known how reversible and how much out of sync the latter process is in relation to the former." I think we know it, see for example ZECMIP study (MacDougall et al., 2020). If CO2 emissions are reduced to zero, atmospheric CO2 concentration will start to

decline, as well as sinks. The warming will stop, but not decrease because of the heat accumulated in the ocean. So on centennial timescale the warming is irreversible.

l.13: "Here we ask three (nontrivial) questions: (1) Can this global-scale memory–Earth's memory–be quantified?" Firstly, you need to define memory and how do you measure it. If it is a memory (timescale) of anthropogenic CO2 in the atmosphere, it is quantified in many studies, see eg Archer et al. (2009) or Joos et al. (2013).

l.14: "(2) Is Earth's memory a buffer which is negligently exploited; and in the case that it is even a limited buffer, what is the degree of exploitation?" I do not understand at all the concept of "exploitation of the Earth's memory". Sounds as you try to formulate a policy-relevant point. However, if you cannot translate it into the terms already accepted in the climate policy framework, such as remaining carbon budget until certain climate target, you miss the policymaker auditorium. See my point above.

l.22: "We find that since 1850, the atmosphere–land/ocean system has been trapped progressively in terms of persistence (i.e., it will become progressively more difficult to train-relax the system)". Again, you need to translate that language ("tapped", "persistence") into more common terminology used in physics and biogeochemistry.

p.3, l.2. "Approximately 60% of Earth's memory had already been exploited by humankind prior to 1959." And what is if it is 100% of memory used? Also, how much could be exploited since last glacial maximum or PETM?

l. 3: "We expect system failures globally well before 2050." What kind of failures are you talking about? Ice sheet collapse, heatwaves, etc.? Be explicit.

Page 4. l.21 to p5., l3. Here is a repetition of the questions from the abstract without any detail why these terms are used and why the questions are important.

Page 5, l. 19: "In view of the aforementioned questions, we chose a rheological stress-strain ($\sigma$-$\varepsilon$) model" I do not see a link between questions and the model choice. I do not see why this formulation is better than a standard climate-carbon box model formulation used for emulators. Is it more universal and less empiric?

p.12, l. 23: the units for NPP are missing, as well as for GPP called global photosynthetic carbon influx in l. 7 on page 13. This renaming of well-known terms and absence of units (eg PgC/yr for NPP, ppmv for the CO2 concentration) in many places is really annoying. Units for most variables are also absent on many plots. This complicates understanding as readers have no idea whether the values are on a global scale, normalized or not, etc.

p.13, l.3: what is "CO2L" – a variable? Multiplication of CO2 and L?

l. 6: what are L units?

l.8: "biomass production" and productivity are different, as there is autotrophic respiration which doesn't turns into biomass. Again, units for beta are missing.

Figs.2, 3, 4,5,6: units are missing on x and y axes.

Fig. 7: what does time in units of 1 means? Is it normalized to some reference time?

Page 29, l.11-12: "They may simply collapse globally". This is a pure speculation. What "they" means here: natural regimes or natural elements (like land or ocean)? How can land or ocean collapse? The carbon uptake would not collapse, but relative uptake fraction will decrease. See Canadell et al. (2021).

l.17: what are "upstream emissions"?

p. 32, l. 1: "land and oceans became less viscous". What do you mean by land viscosity? Viscosity of carbon uptake?

l.6-8: "while its persistence (path dependency) increases by approximately a factor 2–3 if the release of emissions globally continues as before." So what? Are there any consequences for climate if the factor rises up to 10? This should be discussed.

References

Archer, D., et al.,  2009. Atmospheric lifetime of fossil-fuel carbon dioxide, Annual Reviews of Earth and Planetary Sciences, 37, 117-134.

Canadell, J. G., et al. 2021, Global Carbon and other Biogeochemical Cycles and Feedbacks. In: Climate Change 2021: The Physical Science Basis. Contribution of Working Group I to the Sixth Assessment Report of the Intergovernmental Panel on Climate Change, Cambridge University Press. In Press. https://www.ipcc.ch/report/ar6/wg1/#FullReport

Joos, F., et al. 2013: Carbon dioxide and climate impulse response functions for the computation of greenhouse gas metrics: a multi-model analysis, Atmos. Chem. Phys., 13, 2793-2825.

MacDougall, A., et al. (2020). Is there warming in the pipeline? A multi-model analysis of the Zero Emissions Commitment from CO2. Biogeosciences, 17, 2987-3016. doi:10.5194/bg-17-2987-2020

---

## Author Comment (AC1)

Dear Reviewer #1,
Dear Victor Brovkin,

Thank you very much for the review of our Ms (esd-2021-27). We highly appreciate your efforts in terms of time and input! And, to stress, it is a pleasure for us to respond to your numerous comments because this will help to improve our Ms and also uncover many details which underlie our Ms but which we did not mention for the sake of keeping the Ms focused.

In responding, we follow your structure – general comments and specific comments.

**General comments:**

1.  1st para, 1st sentence:

*The manuscript "Quantifying memory and persistence in the atmosphere–land/ocean carbon system"… sticks out of usual approaches to model carbon cycle …*

There exists a misunderstanding. We do not model the carbon cycle. We follow a stress-strain approach—as stated in our very first sentence of the Abstract—not a mass balance approach! The mass-balance based system which you are referring to has units of $PgC\ y^{-1}$, the system within which we are thinking is in units of Pa. Of course, we are extremely happy if the carbon cycle and its various elements are understood internally as consistently as possible because this gives us greater confidence in the parameters which we require / need to estimate (such as the compression and damping characteristics)—but this is not a prerequisite. The stress-strain approach comes with its own, a built-in consistency!

We will come up with appropriate inserts (to both Abstract and Motivation) to stress this nuance / to help avoid this misunderstanding.

2.  1st para, 3rd sentence:

*However, as the field of climate and carbon cycle dynamics is well established and presented by a range of models from zero-dimensional to fully-fledged Earth System models, …*

We fully agree. There exists a wide range of (mass-balance based) carbon-cycle models ranging from simple to complex; however, not, to the best of our knowledge, a global stress-strain model. In retrospect, we know how much complex three-dimensional climate / global change models have benefited from the safeguard of models though simpler but still insightful—such as radiation transfer models, energy balance models, box-type carbon-cycle

models[1] to name a few—but, amazingly, a global stress-strain model is missing in that suite of support models. This is surprising because this model offers further insight into the global atmosphere–land/ocean system (here in terms of three unique parameters—delay time, memory and persistence), especially under favorable conditions. This is why we say on p. 8 (14–17):

*Yet, to the best of our knowledge, a rheological approach to unravel the memory-persistence behaviour of the global atmosphere–land/ocean system in response to the long-lasting increase in atmospheric $CO_2$ emissions had not been applied before.*

and on p. 5 (12–16) explaining, in particular, what we mean by "favorable conditions" ($\rightarrow$ see also our General Responses 6 and 8):

*Under optimal conditions (referring to the long-term stability of the temporal offset), the temporal-offset view even suggests that we can refrain from disentangling the exchange of both thermal energy and carbon throughout the atmosphere–land/ocean system, as it is done in climate-carbon models ranging from simple to complex (Flato et al., 2013; Harman and Trudinger, 2014).*

We suggest addressing this issue together with our insert mentioned under General Response 1 above.

3.  1st para, 3rd and 4th sentence:

*… the authors have to link their definitions and approaches to the ones existing in the field. For example, what does resilience mean in terms of remaining carbon budget until climate goal?*

We wholeheartedly agree—linking *resilience* with the *remaining carbon budget until climate goal* is an interesting issue! However, it is one, as we would argue, which requires a discussion on its own (some would even argue that this issue is not yet fully understood). In any case, going into that issue would not allow us to keep our Ms stringently focused. Let us explain below our thinking behind this statement:

Taking a look at material sciences tells us that resilience is defined as the ability of a material to absorb energy when it is deformed underlined{elastically}, and to release that energy upon unloading. In that systems thinking, resilience comes in units of Pa.[2] However, with reference to our global carbon emissions experiment since 1850 (we comment on other time scales below), we have observed a continuous increase in stress only, underlined{no} relaxation. As a matter of fact, we must even expect a hysteresis effect (see also your comment below) if emissions are decreased and the atmosphere–land/ocean system relaxes. This makes sense because we deal with a system which exhibits memory.
* * *
[1] Cf. also https://eos.org/opinions/when-less-is-more-opening-the-door-to-simpler-climate-models

[2] Cf. https://en.wikipedia.org/wiki/Resilience_(materials_science)

That is, given the global-scale observational data at hand, defining resilience in our systems thinking would be premature. If we would dare attempting a prognostic outlook, we probably would link resilience with hysteresis (if the system would still be capable to relax—which we don't know). Think of a kite with a tail. The longer and heavier the tail, the greater the hysteresis we must expect once we steer the kite into the opposite direction. That is, the "smaller" hysteresis the more resilient the system should be, right?

By way of comparison, in ecosystem sciences resilience is understood as the inherent ability to absorb various disturbances and reorganize while undergoing state changes to maintain critical functions.[3] As a consequence, discussions on measuring resilience focus on time; more specifically, on the speed of recovery after a disturbance.[4]

The discussion so far already demonstrates that it is too early to start a discussion on the similarities and dissimilarities of a stress-strain approach vs a mass balance approach—and to link resilience to the *remaining carbon budget until climate goal*. (To note in passing, to do so demands a separate paper!) We prefer staying focused and facilitating full acquaintance with the unique assets of a stress-strain approach in this Ms. This is also why we have put the mathematics in place.

Finally, a word to the notion of the *remaining carbon budget until climate goal*, frequently referred to in this review. This is a concept very much favored by the global change modeling (and the mass-balance carbon-cycle) community in the context of forecasting exercises. Put simply, upon agreement of a global warming target (e.g., 2 ℃ until 2050), the allowed budget of greenhouse gas emissions can be derived from the (historic and prognostic) scenarios performed with the help of these models; these are scenarios which meet that target. While the cumulative emissions budget approach comes with many benefits (e.g., cumulative emissions scale linearly with the change in global temperature), it must be acknowledged that the targets discussed currently / so far (1.5 or 2 ℃) are purely arbitrary. (To our knowledge, the 2 ℃ target was mentioned first by the developers of the IMAGE I model at the end of the eighties, beginning of the nineties.) However, we are not saying that these targets are unreasonable.

By way of contrast, our stress-strain approach allows coming up with boundary values for the system globally ($\rightarrow$ more specifically, for delay time, memory and persistence; we comment on these boundary values below), if long-term historical conditions are perpetuated (we do not forecast). While the parameter values we use, the details of our experimental setting etc. can all be discussed, the existence of these boundary values cannot be subject to discussion. Their existence is an inherent feature of the stress-strain approach and, consequently, boundary values are not arbitrary!

With these explanations in place, it becomes clear that the stress-strain approach does indeed offer something equivalent to the notion of the *remaining carbon budget until climate goal*.
* * *
[3] Cf. https://www.sciencedirect.com/topics/earth-and-planetary-sciences/ecosystem-resilience

[4] Cf. https://www.nap.edu/read/18387/chapter/5#67

In a strain-strain systems thinking it is the "unexploited" amounts of delay time, memory or persistence given by the "distance" of their values at a given point in time to their respective boundary values in the future (not to be confused with critical thresholds which the system may reach before). We comment on this unique asset in our General Response 10 below. By way of contrast, in our Ms we refer to the "exploited" amounts of delay time, memory or persistence. In Table 1 we report (dimensionless) times when the various ratios $\frac{T}{T_\infty}$ (thus, $\frac{P}{P_\infty}$) and $\frac{M}{M_\infty}$ are 0.5 or 0.95 (to which we refer throughout our Main Findings section; backed-up by our Supplementary Data file, where we give these ratios for annual time steps).

4.   1st para, penultimate sentence:

*What are stress and strain of the Earth system, and how can we measure them.*

Confer Figure 1: In our systems thinking, the stress is given by the anthropogenic $CO_2$ emissions (in units of Pa), while the strain (in units of 1) is given by the expansion of the atmosphere by volume and the uptake of $CO_2$ by sinks; as stated on p. 7 (7–8 ):

*For an observer it is the overall strain response of the atmosphere–land/ocean system (expansion of the atmosphere by volume and uptake of $CO_2$ by sinks) that is unknown.*

On p. 7 (9–14 ), we explain how to get a hold of the strain:

*... since atmospheric $CO_2$ concentrations have been observed to increase exponentially (quasi continuously), the strain can be expected to be exponential or close to exponential. In addition, we provide independent estimates of the likewise unknown compression and damping characteristics of the MB. This a priori knowledge allows equations (1a) and (1b) to be used stepwise in combination to narrow down our initial estimate of the K/D ratio, in particular.*

We suggest a short insert (most likely by expanding figure caption 1) which reiterates / summarizes the known and unknown quantities. We will also edit / replace globally: *MB* [Maxwell body] by *MB representing the atmosphere–land/ocean system*, where appropriate.

5.   1st para, last sentence:

*What does it mean that memory is exploited?*

Memory is defined by equation (4) and scrutinized further in Supplementary Information 3:

$$\sum_{i=1}^{n-1} q_\beta^i = \sum_{i=1}^{n-1}(q_\alpha q)^i = past;$$

that is, as the sum over an exponential ($q$ -weighted) strain which the system had experienced in the past; with $n = today$ and $q = exp\left(-\frac{K}{D}\Delta t\right)$

($\rightarrow$ see also our General Response 8). Equation (4) also tells us that there exists a boundary value for memory—which, as we know, is inherent in the atmosphere–land/ocean carbon system. Hence, our question posed in the Ms whether or not memory is exploited during the

currently ongoing global-scale stress-strain experiment propelled by humankind. We suggest expanding on the notion of "exploitation".

6.  2nd para, last two sentences:

*However, high level of warming was seen in the Earth history many times and the term "stress" is simply not that easy to interpret. I also anticipated that the model could be more universal and be applied to the other periods in the Earth periods such as glacial cycles.*

We selected the time interval 1850–2015 because it appears most appropriate to staying focused and explaining a stress-strain experiment straightforwardly / fully mathematically. This facilitates understanding.

To your question: We consider interpreting "stress" more broadly less of a problem (from an atmospheric concentration point of view); while we think that changes in system characteristics in time are the real problem and are more difficult to capture (what we work on right now). Since 1850 the Earth system was increasingly anthropogencially forced; with the consequence that system characteristics could not change as quickly as they would have done in the case of a slow(er) forcing / more time available. (Note that we proceeded time-wise in intervals during which we keep system characteristics constant and thus can accommodate with their changes over time.) That is, the speed of forcing comes in crucially.

In brief: The potential of applying a stress-strain approach universally / under other conditions exists but still requires further scientific thinking. However, we are optimistic that this will be possible at some time in the future.

7.  3rd para, 1st sentence:

*I also expected that the model could explain fundamental features of the carbon system, for example, why the combined anthropogenic carbon uptake by land and ocean today is about 60%, and not 40% or 80%.*

Sorry, but this is a problem for the mass-balance carbon-cycle community!

8.  3rd para, 2nd sentence:

*Unfortunately, the model parameters are tuned to reproduce observations, assuming that pre-industrial state is without stress.*

See our General Response 4 above wrt how we operate equations (1a) and (1b) in combination. Also note that we tune skillfully: we tune within parameter limits (for K and D) which we derive independently!

Correct, we assume that the pre-industrial state is without (anthropogenic) stress. (For clarification: we considered initial zero-stress conditions increasingly backward in time—in 1959, 1900 and 1850; and non-zero stress conditions in 1959 if our stress-strain experiment starts prior to 1959.) But this comes without consequences for memory. On p. 9 (16–17) we state

*Delay time, memory, and persistence are characteristic of the MB and are defined independently of initial conditions.*

and in experiments B and C we show that $M_\infty$, the limit of memory (here with reference to 1959), ranges narrowly between 19–22 in units of 1 (see p. 30 [10] and also Table SI10-2 in our Supplementary Information 10); supporting the above statement that initial conditions do not matter for deriving memory.

This is in line with how mathematics works: memory starts accumulating from year 1 onward on the dimensionless time scale  (memory is not defined for $n = 0$), meaning that memory refers to the current dynamics of the system, not to pre-initial conditions / times. This also makes sense physically. What would be the alternative? That memory goes back to when Earth was created? This would only be so if $n = 0$ would specify the time of the formation of Earth.

9.  3rd para, last two sentences:

*Actually, most of the time during the last million years the Earth System was in cold conditions under low CO2 concentrations. If one uses the Last glacial maximum ca. 20 thousand years ago as a reference, would the pre-industrial state be a stress for the Earth System?*

See our previous response and General Response 6 above.

10. 4th para, as a whole:

*In summary, using the new approach such as rheological model is useful in case the method brings something new comparing to the current set of models used for policy applications. However, I just do not see what new can we learn from this approach. It looks more as reformulation of the problem in new variables which are non-measurable and have little content to be interpreted by policy makers, such as remaining carbon budget. I think the approach is interesting, so I do not recommend to reject the manuscript, but without substantial rewriting and interpreting terms in measurable quantities this study will miss the auditorium.*

To reiterate, we apply a stress-strain approach to simulate the effect of the continued release of $CO_2$ emissions into the atmosphere. The (known) stress is given by $CO_2$ emissions from fossil fuel burning and land use; the (unknown) strain response of the atmosphere–land/ocean system is given by the expansion of the atmosphere by volume and the uptake of $CO_2$ by sinks. The importance here is not that this approach can be refined from a rheological perspective but that it allows system characteristics—here the system's compression and damping characteristics (and also the altitude of its top of the atmosphere)—to be derived; these are characteristics that any complex, geographically explicit model (e.g., a general circulation model  or global vegetation model) should exhibit if it were globally averaged. The great advantage of simplified, physically constrained models being calibrated to global-scale observations is that they can then be formulated to replicate the globally averaged

behavior of complex models—or even help to readjust them and put them on a globally more realistic track. To the best of our knowledge, a stress-strain model was missing in that suite of simplified models.

At this point of time, the primary target audience of our Ms are Earth system / modeling experts (which is also why we describe our stress-strain approach in full mathematical detail), and other experts (including decision-makers) with some mathematical-physical background. As a matter of fact, our Ms comes with real-life consequences. Our stress-strain approach suggests that the timeframe for action to curb $CO_2$ emissions at the political level must be shortened—considerably. At a deeper level, with our approach being stress-strain constrained, it obeys consistency (unlike prospective models in general). A direct consequence is that the atmosphere–land/ocean carbon system will lose its ability to build up memory effectively well before 2050—which we understand as a limit on the system to respond while still within its own natural regime. We consider this finding to be just as alarming as the combined outcomes of complex global-change models (based on prospective changes in temperature, precipitation, etc.).

Last, but not least, our stress-strain approach offers an alternative avenue to monitoring environmental change. It allows system characteristics (delay time, memory, and persistence) to be quantified if historical conditions were to continue as in the past. That is, our approach allows changes in these system characteristics to be calculated against their boundary values, whereas environmental (e.g., ecosystem) researchers typically refer monitored changes to initial (e.g., pre-industrial) conditions. We anticipate that merging the two scales of change will provide new insights into identifying tipping points and system failures.

To summarize, the purpose of our extensive response here is to stress (again) that (i) we must strike a balance between demonstrating a workable approach on the one hand and, on the other hand, taking on board additional information to the extent needed / justified; and that (ii) we interpret the request of rewriting the Ms as making sure that potential misunderstandings (as those described / identified before) are avoided and that a reader is not misled anywhere, while ensuring that the workability of our approach is not impaired. We suggest that we go, after having implemented all changes suggested before, through the Ms again to scrutinize where additional strategic improvements in that direction can be done.

**Specific comments:**

1.  P.2, l.6:

*"**the CO2 concentration in the atmosphere increases (rather quickly).** Concomitantly, the atmosphere warms and expands, while **part of the carbon is locked away (rather slowly) in land and oceans**, likewise under the influence of global warming."* This statement is in disagreement with data. Over last decades, airborne fraction of CO2 emissions is rather constant, see for example Chapter 5 in IPCC AR6 WGI (Canadell et al., 2021). That means that $CO_2$ in atmosphere is increasing as fast as sinks on land or in the ocean.

No, there is no disagreement! We see an equivalent effect: see p. 30 (4–6):

*Between 1850 and 1959–2015 (i) the compression modulus $K$ increased from ~2 to 10–13 Pa (the atmosphere became less compressible) while (ii) the damping constant $D$ decreased from ~468 to 459–462 Pa y (the land and oceans became less viscous) ...*

A note in passing: in our systems thinking, the airborne fraction is less relevant because it is neither stress nor strain; but we are aware that this parameter is used in the mass-balance carbon-cycle community.

2.  l.11:

*"It is not known how reversible and how much out of sync the latter process is in relation to the former."* I think we know it, see for example ZECMIP study (MacDougall et al., 2020). If CO2 emissions are reduced to zero, atmospheric CO2 concentration will start to decline, as well as sinks. The warming will stop, but not decrease because of the heat accumulated in the ocean. So on centennial timescale the warming is irreversible.

We are aware of this purely prognostic multi-model comparison study. To reiterate: We do not forecast, nor do we speculate about whether or not sinks strain-relax at all and, if so, about the hysteresis involved if emissions, from a specified time onward, are set to zero or decline gradually.

However, we note that we quantify, in our stress-strain context of increasing emissions, the two relevant times which also underly the afore-mentioned multi-model study—see p. 9 (20–24):

*... where $\beta = 1 + \frac{D}{K}\alpha$ and $q_\beta^t = exp\left(-\frac{K}{D}\beta t\right)$. The term $\frac{D}{K\beta}$ represents a time characteristic of the MB under (here) exponential strain (i.e., of the MB that responds to the stress acting upon it), whereas $\frac{D}{K}$ is the relaxation time of the MB (i.e., of the MB that relaxes unhindered after the stress causing that strain has vanished, or that responds to strain held constant over time; also known as the relaxation test ...*

3.  l.13:

*"Here we ask three (nontrivial) questions: (1) Can this global-scale memory−Earth's memory−be quantified?"* Firstly, you need to define memory and how do you measure it. If it

*is a memory (timescale) of anthropogenic CO2 in the atmosphere, it is quantified in many studies, see eg Archer et al. (2009) or Joos et al. (2013).*

We define memory → see our General Response 5. (Note: all quantities to determine memory are known!) Our definition of memory refers to the exponential ($q$ -weighted) strain which Earth had experienced in the past; it does not refer to the time scale of anthropogenic $CO_2$ in the atmosphere. We will come up with appropriate inserts / within-Ms references to improve clarity and avoid potential misunderstandings.

4. l.14:

*"(2) Is Earth's memory a buffer which is negligently exploited; and in the case that it is even a limited buffer, what is the degree of exploitation?" I do not understand at all the concept of "exploitation of the Earth's memory". Sounds as you try to formulate a policy-relevant point. However, if you cannot translate it into the terms already accepted in the climate policy framework, such as remaining carbon budget until certain climate target, you miss the policymaker auditorium. See my point above.*

We already explained that memory comes with a boundary value and that it is justified to speak of "exploitation" → see our General Responses 5 and 3. We suggest expanding on the notion of "exploitation".

We also already explained that, if a system loses its ability to build up memory effectively, this is understood as a limit on the system to respond within its own natural regime → see our General Response 10.

We also responded already to the notion of a *remaining carbon budget until certain climate target* → see our General Response 3.

5. l.22:

*"We find that since 1850, the atmosphere–land/ocean system has been trapped progressively in terms of persistence (i.e., it will become progressively more difficult to* [s]*train-relax the system)". Again, you need to translate that language ("*t[r]apped*", "persistence") into more common terminology used in physics and biogeochemistry.*

We define persistence in equation (5). This equation also tells us that there exists a boundary value for persistence (which we specify for each experiment → see, e.g., Table 1). To facilitate understanding, we explain that persistence can be understood as path dependency; see p. 11 (7–11):

*Equation (5) can be shortened to $T{\cdot}P=1$. If we assume that $q$ can be changed in retrospect at $n=0$, this equation tells us that if $T$ —that is, $\Delta M$ per $\Delta q$ (or, likewise, $\Delta M/M$ per $\Delta q/q$; see the first part of equation [3])—is small, $P$ is great because the change in the system's characteristics (contained in $q$) hardly influences the MB's past, with the consequence that the past exhibits a great path dependency, and vice versa.*

We'd argue that "path dependency" is generic and understood across scientific communities. This allows persistence to be understood immediately in relative terms (greater persistence = greater path dependency; and vice versa).

We'd also argue that our stress-strain terminology is already in line with physics—which we prefer as reference, especially since our Ms cuts across various scientific disciplines. It must therefore be expected that a reviewer from another background may request adjusting the physical terminology used by us to the terminology favored in his / her scientific community. We suggest strengthening our physical language even more by way of an appropriate (clarifying) insert. An alternative could also be a globally relevant insert in the beginning of our Ms which informs a reader that our terminology is influenced by physics, more specifically, by the terminology used within rheology.

6. p.3,l.2:

*"Approximately 60% of Earth's memory had already been exploited by humankind prior to 1959." And what is if it is 100% of memory used? Also, how much could be exploited since last glacial maximum or PETM?*

See p. 30 (9–10) (and also Table SI10-2 in the Supplementary Information 10):

*... while (v) memory $M_\infty$ decreased from ~52* [equivalent to 100%] *to 19–22 on the dimensionless time scale.*

Wrt the last glacial maximum or PETM → see our General Response 6.

7. l.3:

*"We expect system failures globally well before 2050." What kind of failures are you talking about? Ice sheet collapse, heatwaves, etc.? Be explicit.*

Our stress-strain model is global → see our General Responses 2 and 10 (that is, we cannot go sub-global). We will make clear that *failures* is to be understood with reference to the entire system, as used in rheology → see also Specific Response 5: An alternative could also be a globally relevant insert in the beginning of our Ms which informs a reader that our terminology is influenced by physics, more specifically, by the terminology used within rheology.

8. Page 4. l.21 to p5, l3:

*Here is a repetition of the questions from the abstract without any detail why these terms are used and why the questions are important.*

Our very first ESD reviewer had requested us to repeat these questions in the introduction—which we think makes sense because they are the red thread running through our Ms.

9. Page 5, l. 19:

*"In view of the aforementioned questions, we chose a rheological stress-strain ($\sigma$-$\varepsilon$) model" I do not see a link between questions and the model choice. I do not see why this formulation is better than a standard climate-carbon box model formulation used for emulators. Is it more universal and less empiric?*

We think in a stress-strain context → see our General Response 1.

Rheology  is the study of the flow of matter under conditions in which they respond with plastic flow rather than deforming elastically in response to an applied force. Once one begins thinking in a stress-strain context, it becomes immediately obvious to explore the advantages of a rheological approach (→ see our General Response 10). Above and beyond, "memory" is known in rheology.

We suggest an appropriate insert to facilitate directing a reader.

10. p.12, l. 23:

*the units for NPP are missing, as well as for GPP called global photosynthetic carbon influx in l. 7 on page 13. This renaming of well-known terms and absence of units (eg PgC/yr for NPP, ppmv for the CO2 concentration) in many places is really annoying. Units for most variables are also absent on many plots. This complicates understanding as readers have no idea whether the values are on a global scale, normalized or not, etc.*

We acknowledge that our Supplementary Information (https://doi.org/10.22022/em/06-2021.123 → see Data Availability section in the Ms) went unnoticed by the reviewer. Our Supplementary Information lists at its end all acronyms (including units where appropriate) used in the Ms → see *Acronyms (used in Ms No. esd-2021-27 and in this SI)*. We will take this list out the Supplementary Information and insert it into our Ms where we place it at the end (before References).

We will also check where (i) units are not mentioned in figure captions (preferred over mentioning units in figures) or (ii) are mentioned over-frequently; while paying attention to the fact that our Ms must also satisfy readers from scientific communities other than the mass-balance carbon-cycle community.

11a.  p.13, l.3: *what is "CO2L" – a variable? Multiplication of CO2 and L?*

11b.  l. 6: *what are L units?*

11c.  l.8: *"biomass production" and productivity are different, as there is autotrophic respiration which doesn't turns into biomass. Again, units for beta are missing.*

11d.  Figs.2, 3, 4,5,6: *units are missing on x and y axes.*

Will also / automatically be taken care of → see our Specific Response 10.

12. Fig. 7:

*what does time in units of 1 means? Is it normalized to some reference time?*

This we explain in our Ms on p. 10 (1–4):

*However, to ensure that exponents still come in units of 1 after we split them up, we introduce the dimensionless time $n = \frac{t}{\Delta t}$ globally (which will be discretised in the sequel when we refer to a temporal resolution of 1 year and set $\Delta t = 1y$), such that, for example,*

$$q^t = exp\left(-\frac{K}{D}\Delta t\right)^n.$$

That is, $n$ refers to a dimensionless time scale given in year-equivalent units. A note in passing: this mathematical hint—that exponents must always come in units of 1 when split up—is a precious gift! It helps putting our system parameters delay time, memory and persistence on a solid footing!

13. Page 29, l.11-12:

*"They may simply collapse globally". This is a pure speculation. What "they" means here: natural regimes or natural elements (like land or ocean)? How can land or ocean collapse? The carbon uptake would not collapse, but relative uptake fraction will decrease. See Canadell et al. (2021).*

As a matter of fact, this is fully explained in the context of the entire para → see p. 29 (8–12):

*Finally, it is important to note that it is prudent to expect that natural elements (like land and oceans) will not continue to maintain their damping capacity—or their capacity to embark on a, most likely, hysteretic downward path in the case of a sustained decrease in emissions—even well before they reach the limits of their natural regimes. They may simply collapse globally.*

However, we will replace the last sentence by *These elements may simply collapse globally, i.e., from a holistic systems perspective.*

Wrt to "collapse": this term refers to state changes / what rheology is about → see our Specific Response 9. We suggest an appropriate insert for clarification → see also Specific Response 5: An alternative could also be a globally relevant insert in the beginning of our Ms which informs a reader that our terminology is influenced by physics, more specifically, by the terminology used within rheology.

Wrt continued carbon uptake: Your statement appears bold / speculative (at least on short time scales). Look at what is currently happening at the Marmara Sea![5]
* * *
[5] https://www.downtoearth.org.in/news/environment/turkey-should-mitigate-factors-aiding-growth-of-sea-snot-in-sea-of-marmara-expert-77741

14. l.17:

*what are "upstream emissions"?*

= prior emissions. "Upstream" is a common term used by emission experts. We will explain
or replace this term.

15. p. 32, l. 1:

*"land and oceans became less viscous". What do you mean by land viscosity? Viscosity of
carbon uptake?*

Admittedly, we use rheological terminology, but the meaning becomes clear in the context of
the entire sentence → see p. 31 (22–24) to p. 32 (1–2 ):

*However, the overall effect of the continued release of GHG emissions since 1850 on the
K/D ratio is unambiguous—the ratio increased by a factor 4–6 (K increased: the
atmosphere became less compressible; D decreased: land and oceans became less ==viscous==),
resulting in the aforementioned changes in delay time, memory, and persistence.*

That is, we talk about the damping characteristic *D* resulting from the carbon uptake of land
and oceans but with reference to what that means for the overall behavior of land and oceans
in a stress-strain context (→ see our Figure 1: *D* describes how the damping element behaves
depending on an applied stress).

Will also / automatically be taken care of → see our Specific Response 5.

16. l.6-8:

*"while its persistence (path dependency) increases by approximately a factor 2–3 if the
release of emissions globally continues as before." So what? Are there any consequences for
climate if the factor rises up to 10? This should be discussed.*

This (final) sentence summarizes an aspect of our Main Findings in a wider context → see p.
30 (8–10).

We already explained that there exists a boundary value for persistence → see our Specific
Response 5.

Persistence can be understood parallel / 1:1 to memory → see our Specific Response 4:
While the system loses its ability to build up memory effectively under continued stress (→
to be understood as a limit on the system to respond within its own natural regime), the
system's path dependency increases, meaning that that the system is increasingly locked in.

We will insert an appropriate sentence / explanation in the Main Findings section.

Again, with great thanks to you for having reviewed our Ms,
along with our sincere greetings,

Matthias Jonas, Rostyslav Bun, Iryna Ryzha and Piotr Żebrowski

---

## Author Comment (AC2)

**Note to ESD's Editorial Support Team:**

We do not yet revise our Ms, as instructed by your email dated 21 September 2021. Nonetheless, in our response below to Reviewer #2 we indicate which revisions we envisage. These will be done in a track-change mode to facilitate making decisions about the further handling of our Ms.

Dear Reviewer #2,

Thank you very much for the review of our Ms (esd-2021-27). We highly appreciate your efforts in terms of time and input! And, to stress, it is a pleasure for us to respond to your valuablecomments because this will help to improve our Ms and also uncover the details which underlie our Ms but which we did not mention for the sake of keeping the Ms focused.

In responding, we follow your general comments.

**General comments:**

1st para, last sentence:

*Particularly, the readability needs to be improved.*

This we aim at as well → see our responses below.

1. General Comment 1:

*One of my main concerns is about the stress-strain model. Why is this model useful in understanding the memory/persistence in the system? More detailed discussions and arguments are needed.*

This question is best answered by decomposing it wrt two insights:

(1) Rheology is the study of the flow and deformation of matter reflecting the interrelation between force, deformation and time.[1] Once one begins thinking in a stress-strain (modeling) context, it becomes immediately obvious to explore the advantages of a rheological approach, an important, if not the most important, one being consistency. Above and beyond, "memory" is known in rheology, as is "hysteresis" (not so "persistence").[2,3]

(2) However, the usefulness of a stress-strain approach achieves its full potential only in combination with the insight that (i) mathematics offers a direct (and quantifiable) handle on delay time, thus on memory and persistence; and that (ii) the emergence of memory is contained in each ordinary differential equation describing processes which come with a lag-
* * *
[1] Cf. also https://cdn.technologynetworks.com/TN/Resources/PDF/WP160620BasicIntroRheology.pdf

[2] Cf. also https://www.itcp.kit.edu/wilhelm/download/Introduction%20to%20Rheology_2019.pdf

[3] Cf. also https://metalurji.mu.edu.tr/Icerik/metalurji.mu.edu.tr/Sayfa/PlasticityW9.pdf

time —as it is the case with a stress-strain approach in the context of a system exhibiting memory. That is, the definitions of memory and persistence are not arbitrary; they are a logical consequence of how processes with a lag time are described in physics traditionally.

That is, the combination of insights 1 and 2 offers the potential of exploring four assets in an Earth systems context: consistency, delay time, memory and persistence—what had not been done before.

Still, however, to do so requires, in addition, the workability of this endeavor / our global atmosphere–land/ocean carbon systems approach to be ensured. This is why we say, e.g., on p. 5 (12–16):

*Under optimal conditions (referring to the long-term stability of the temporal offset), the temporal-offset view even suggests that we can refrain from disentangling the exchange of both thermal energy and carbon throughout the atmosphere–land/ocean system, as it is done in climate-carbon models ranging from simple to complex (Flato et al., 2013; Harman and Trudinger, 2014).*

(There exist more favorable conditions → see also our response to Reviewer #1.)

In brief, this is what our Ms is all about—it  demonstrates the workability of a stress-strain approach under favorable conditions and in an Earth systems context! We can only speculate that the afore-mentioned circumstances explain why a stress-strain approach had not been applied so far.

We suggest complying with your request of providing additional background knowledge along these lines (without adding more mathematics).

2.   General Comment 2:

*The "Abstract" should be improved. Now it is quite similar to the first three paragraphs of the "Motivation".*

This can certainly be done.

3.   General Comment 3:

*A more detailed overview of the previous progresses on climate persistence may be helpful. In the current version, the authors only mentioned the previous studies very briefly, as shown on page 7 (lines 24-25), page 8 (lines 1-4). However, what are the limitations of the previous studies? I would suggest the authors to make a more detailed introduction.*

We will extend our Motivations section accordingly.

Wrt an overview of "persistence": As a matter of fact, we have done a survey on how "memory", "persistence" and other / equivalent terms are understood across climate studies, economics and finance, and geophysics and physics. It should not come as a surprise that, therein, these terms are generally "defined" statistically, based on correlation principles. By way of contrast, definitions of memory and persistence are coming as quantifiable system

parameters / for free under a stress-strain approach (→ see also General Responses 2 and 4). This was the main reason why we were overly brief on "memory" and "persistence" in our Ms.

4.  General Comment 4:

*The definitions of "delay time", "memory", and "persistence" should be clearly given in the main text. What is the differences between "memory" and "persistence"?*

We defined delay time, memory, and persistence in equations (3) to (5) in the Methods section. However, we suggest referring to these equations when they are used in the Main Findings section, where we will also embed them also more clearly in a physical context.

To the difference between memory and persistence, in particular:

Memory is given by

$$M(q, n) = \frac{1 - q_\beta^n}{1 - q_\beta} = \sum_{i=0}^{n-1} q_\beta^i = \sum_{i=0}^{n-1} (q_\alpha q)^i = past;$$

that is, as the sum over an exponential ($q$ -weighted) strain $q_\alpha$ which the system had experienced in the past; with $n = today$ and $q = exp\left(-\frac{K}{D}\Delta t\right)$.

By way of contrast, persistence is given by

$$P(q, n) := \left(\frac{\frac{1}{M}}{\frac{1}{q_\beta}} \frac{\partial M}{\partial q_\beta}\right)^{-1};$$

that is, as the inverse of the (normalized) underline{derivative of M} (i.e., over the entire past!) by $q_\beta$, the meaning of which we explained on p. 11 (7–11 ):

*Assuming that q can be changed in retrospect at $n = 0$ [while a is held constant], this equation tells us that, if ... $\frac{\Delta M}{M}$ per $\frac{\Delta q}{q}$ ... is small, P is great because the change in the system's characteristics (contained in q) hardly influences the MB's past; with the consequence that the past exhibits a great path dependency; and vice versa.*

We will try to carve out also this difference more clearly in physical terms.

5.  General Comment 5:

*Regarding the "Data and Conversion Factors" section, I would suggest the authors to add more information (about the data used in this study) here. It is not convenient for the readers to search for the data information in the supplementary information. At least, some basic information should be provided in the main text.*

This can certainly be done.

6.  General Comment 6:

*When estimating the compression modulus K, why is the atmosphere assumed to be represented by a Hooke element in the MB? What is a Hooke element?*

A Hooke element is a linear elastic body (Hooke body, Hooke model, Hooke element, elastic spring). It represents the behavior of a perfectly elastic (lossless) material. Stress is proportional to strain.[4] Hence, our careful note on p. 19 (7–10):

*Compared to the slow uptake of carbon by land and oceans, we assume the atmosphere to be represented well by a Hooke element in the MB and this to serve as a (sufficiently stable) surrogate physical descriptor for the reaction of the atmosphere as a whole (Sakazaki and Hamilton, 2020).*

7.  General Comment 7:

*The current manuscript is very technical. For me, I would like to see more explanations of the results from the perspective of climate sciences.*

This can certainly be done (while keeping our Ms stringently focused). We can think of an additional section at the end of our Ms where we can hint at the value of a global stress-strain model as a support model for the support of complex, geographically explicit models (e.g., general circulation or global vegetation models) → see also our response to Reviewer #1).

Again, with great thanks to your very constructive comments,
along with our sincere greetings,

Matthias Jonas, Rostyslav Bun, Iryna Ryzha and Piotr Żebrowski
* * *
[4] Cf. http://www.earthphysics.sk/mainpage/stud_mat/Moczo_Kristek_Franek_Rheological_Models.pdf

---

## Author Response (AR2)

Dear Reviewer,

Dear Prof. Zhengui Xie,

Thank you very much for the review of our Manuscript (**Ms** esd-2021-27). We highly appreciate your efforts in terms of time and input!

**Your general comment concerning minor revision:**

1. *The revised manuscript addressed the comments from the two* [previous] *reviewers. However, the sentences Line 16-23 in P2 in the abstract was also described in the section 1 Line 17-25 in P7, please modify the* [paragraph] *or removed in the abstract section.*

We modified the paragraph (see marked Ms).

Please note

- that we also took care of some minor edits throughout the Ms (all marked).

- that we also followed ESD's instructions of how to list references. (To facilitate easy reviewing, these changes are not in a track-change mode.)

- that our Supplementary Material consists of two parts: (i) Supplementary Information (**SI**) and (ii) Supplementary Data (**SD**). The two parts are publicly available at https://doi.org/10.22022/em/06-2021.123.

- that we submit, for your convenience, one PDF file which contains all track-change edits of both Ms and SI. (While the SI still contains tables with colored cells, the Ms does not.)

- that our clean Ms comes as a LaTeX file.